



# Last Interglacial sea-level data points from Northwest Europe

Kim M. Cohen[1], Víctor Cartelle[2], Robert Barnett[3], Freek S. Busschers[4], Natasha L.M. Barlow[2]

[1]Department Physical Geography, Utrecht University, PObox 80.115, 3508TC Utrecht, Netherlands
[2]School of Earth and Environment, University of Leeds, Woodhouse Lane, Leeds LS2 9JT, United Kingdom
[3]Department of Geography, University of Exeter, Rennes Drive, Exeter EX4 4RJ, United Kingdom
[4]TNO Geological Survey of the Netherlands, PObox 80.015, 3508TA Utrecht, The Netherlands

*Correspondence to*: Kim M. Cohen (k.m.cohen@uu.nl)

**Abstract.** Abundant numbers of sites and studies exist that document the Last Interglacial (Eemian, Ipswichian, MIS 5e) coastal record for geographically and geomorphologically diverse NW Europe. This paper documents a database of 141
known Last Interglacial sea-level indicative data points from in and around the North Sea (35 entries in Netherlands, 10 Belgium, 16 in Germany, 17 in Denmark, 8 in Britain) and the English Channel (28 entries for British and 25 for the French side, 3 on the Channel Isles), believed to be a representative and fairly complete inventory and assessment coming from some 80 published sites. The good geographic distribution (some 1500 km SW-NE) across the near field of the Scandinavian and British Ice Sheets and the attention paid to absolute and relative age control are assets of the NW European database
compilation. The research history of Last Interglacial coastal environments and sea-level position for this area is long, methodically diverse and spread over regional literature in several languages. Last Interglacial high-stand shorelines of Dutch and German Bight parts of the North Sea, were of lagoonal and estuarine type and have preserved subsurface (data entry included estimates of non-GIA vertical land motion). In contrast, Last Interglacial high-stand shorelines along the English Channel are encountered above modern sea-level (data entry includes datum definitions). Our review and database
compilation effort drew from the original regional literature, and paid particular attention to distinguishing between sea-level index points (SLIPs) and marine and terrestrial limiting-points.

This paper describes the dominant sea-level indicators produced from region to region, compliant to the database structure of the special issue (WALIS), referenced to original source data. The sea level proxies in majority are obtained from localities
with well-developed lithostratigraphic, morpho-stratigraphic and biostratigraphical constraints. Amino-Acid Racemization information is also prominent, especially in Britain, albeit for many sites the older, lesser quality applications of that technique. The majority of European continental sites have chronostratigraphic age-control, notably through regional Pollen Association Zones of known durations. This greatly helps to separate transgression, highstand ('stillstand') and regression subsets from within the interglacial, useful when summarizing and/or querying the dataset. In all regions, many SLIPs and
limiting points have further independent age-control from luminescence (IRSL, OSL, TL), U-series and ESR dating techniques. Main foreseen usage of this database for the near field region of the European ice sheets is in GIA modelling.



## 1 Introduction

Near-field records of Last Interglacial (LIG; 130-116 ka) sea level, proximal to the location of palaeo ice sheets, are critical for establishing improved reconstructions of past ice sheets, constraining models of solid Earth processes and fingerprinting
the source of ice sheet melt (Dutton et al., 2015; Long et al., 2015). However, the near-field LIG sea level has received comparable little attention compared to the far-field due to the challenges of dating estuarine sequences and the complications of regional glacial isostatic adjustment (GIA). The main aim of this paper is to describe a standardized database of geological sea-level proxies, compiled using the tools available through the World Atlas of Last Interglacial Shorelines (WALIS) project for north west (NW) Europe, in particular around the North Sea and English Channel region.
This is a location that is proximal an extensive MIS 6 Eurasian ice sheet (e.g. Svendsen et al. 2004; Ehlers and Gibbard, 2004; Lambeck et al., 2006; Lang et al., 2018) and has a long history of Quaternary palaeoenvironmental research (e.g., Dixon, 1850; Harting, 1874).

The thickness and nature of the LIG coastal geomorphology and sedimentary sequences in NW Europe varies considerably.
In the North Sea coastal region (e.g. Netherlands, offshore in the North Sea, NW Germany and SW Denmark), key sites are meter-thick infills of topographic depressions in deglaciated terrain, between the maximum ice-margins of the Saalian glaciation (MIS 6) and that of the Last Glacial (Fig. 1) (e.g. Zagwijn, 1983; Streif, 2004; Beets et al., 2005; Konradi et al., 2005). Along the English Channel many LIG sites comprise of flights of raised beaches, e.g. West Sussex coastal plain, southern England (Bates et al., 2010) and Cotentin, northwest France (Coutard et al., 2006). Both along the North Sea and
along English Channel, the mouths of rivers record transgressed palaeovalleys and provide opportunities to constrain the regional LIG sea-level highstand (e.g. Antoine et al., 2007; Briant et al., 2012; Bogemans et al., 2016; Peeters et al., 2016).

Due to the extensive history of palaeoenvironmental research in this region, many of the sites documented here were studied 50+ years ago, by experts in microfossil and sedimentological analysis rather than with a focus on establishing sea level
index points (SLIPs) under now well-developed frameworks (Rovere et al., 2016; van de Plassche, 1986; Hijma et al., 2015). That wealth and history of research is both its strength and a challenge when compiling a database for NW Europe's North Sea and English Channel coastlines.

**Figure 1 (*next page*)  Overview of study area with data points (legend groups cf. Section 5) and Setting / Palaeogeographical**
**context. North Sea and English Channel coastlines and lower reaches of main rivers depict situation at Last Interglacial highstand (newly compiled as part of data screening effort, matching regional reconstructions provided in main SLR data source papers). Ice-limits for the penultimate and last glacial cycles in area of LIG data points similarly compiled from regional studies (Ehlers et al., 2004; Busschers et al., 2008; Moreau et al., 2012; Lang et al., 2018; Gibbard et al., 2018; Cartelle et al., 2021) and adjoined to superregional overviews (Ehlers and Gibbard, 2004; Batchelor et al., 2019). Pleistocene depocentre in North Sea indicated: see**
**Section 4.3. Axis used to arrange presentation in Figs. 5 and 6 indicated. Selection of topographic names from text included, offshore sites with informal short ids from source papers. Bathymetry and DEM backdrop: EMODnet Bathymetry Consortium (2020), their WMS-service which incorporates land data © OpenStreetMap contributors 2020, distributed under the Open Data Commons Open Database License (ODbL) v1.0.**





**Table 1 Global and regional time division schemes**

| *Worldwide division schemes* | | | | | *Regional division schemes* | | |
|---|---|---|---|---|---|---|---|
| *Generic* | *Chrono-stratigraphic* | *Sequence Stratigraphic* | *Marine Isotope Stratigraphy* | | *Mainland NW Europe* | *PAZ subdivision, with varve count durations NW Germany* | *British Isles* |
| (Holocene) | Holocene | Highstand | MIS 1 | | (Holocene) | … | (Holocene) |
| | | *transgression* | *Termination I* | *14.7 ka* | | Lateglacial | |
| Last Glacial | | Lowstand | MIS 2 | LGM | | *~29-27, ~24-21 ka #* | |
| | | | | | | Pleniglacial | |
| | | | MIS 3 | | | stadials, interstadials | |
| | Late Pleistocene | | MIS 4 | | Weichselian | *~70 ka* | Devensian |
| | | Falling stage (w large oscillations) | MIS 5c-a | | | Early Weichselian | |
| Last Interglacial | | | MIS 5d | *~115 ka* | | interstadials, stadials EW | |
| | | *regression* **Highstand** *transgression* | MIS 5e | | Eemian *~125 ka* | E6    ~ 4000 yr E5    ~ 4000 yr E1-E4    ~ 3000 yr | Ipswichian *~125 ka* |
| | | | *Termination II* | *~130 ka* | | LS | |
| Penultimate Glacial | | Lowstand | MIS 6 | PGM | Saalian | *Warthe substage  ~140 ka ##* *Drenthe substage ~160 ka ###* | Wolstonian |
| … | Middle Pleistocene | … | … | | | | |
| | | (prior cycles) | | | | (prior interglacials, glacials) | |

Schemes follows Cohen & Gibbard, 2019. The palynological subdivision column adds E1-E6 durations of Müller (1974), Zagwijn (1983, 1996).

\*   Termination midpoints are MIS stage boundary (Shackleton 1969; Lisiecki & Raymo, 2005; Railsback et al. 2015).

#   Last Glacial glaciation of the study area reaching maxima within MIS 2 and MIS 4. Limits over NE Germany, Denmark and central North Sea (Fig. 1)

##   Warthe substage, with Scandinavian limit just beyond Last Glacial ones in German Bight and onshore (Fig. 1). Offshore landforms LIG transgressed.

###   Drenthe substage, with Scandinavian limit furthest SW across North Sea, Netherlands and Germany onshore (Fig. 1). Offshore landforms LIG transgressed.

The paper covers the study area region-by-region, starting with the North Sea coast of the Netherlands, Belgium, the German

Bight, Denmark and the Danish-German SE of the Baltic sea (compiled by KMC), then the British North Sea coast (Thames Estuary, East Anglia and North England; compiled by NLMB) and offshore records in the British North Sea (complied by





VC), then the British (Southern and Southwest Britain; compiled by RB) and French (Normandy and Brittany; compiled by VC) sides of the English Channel. In NW Europe, age control on LIG estuarine deposits typical relies on relative dating based on micro- and macro-fossil biostratigraphy, and amino-acid racemisation (AAR) of fossil material due to the nature of

the fine-grained silt, clay and organic sediments. More recently studied, or revisited, sites may have absolute OSL or U-series ages where the stratigraphy permitted. Differences in geological-geomorphological settings along and between these large stretches of coast, tabulated in Table 2, have influenced depth of preservation, surveying and mapping strategies, preservation and taphonomy characteristics of typical sites and so on, and hence are touched upon repeatedly in this research history and dataset documentation (mostly in sections 2 and 5). In turn, this strongly influenced our region-by-region

strategies for WALIS data entry (described in sections 3 and 4). For example, where regionally extensive Saalian glaciation landforms or tills are identified (Fig. 1), this identifies overlying marine deposits as Last Interglacial directly (areas with 'paraglacial' geomorphic control; Table 2). Away from the MIS-6 glaciated areas, however, this benefit is lost and identifying depositional record from the LIG can become ambiguous. The geographical differences also echo through in the discussion of NW Europe LIG sea-level data quality (age control, vertical control) towards the end of the paper.


**Table 2: NW Europe WALIS data point totals, split by region.**

| Region | Quaternary Terrain | Main Age control | Last Interglacial | | | Older Interglacials | | |
|---|---|---|---|---|---|---|---|---|
| | | | SLIPs | Mar. limit. | Ter. limit. | SLIPs | Mar. limit. | Ter. limit. |
| *North Sea* | | | | | | | | |
| N Netherlands *Central NL lagoon Rhine estuary* | Saalian (MIS 6) paraglacial*. Substrate is further Quaternary sedimentary basin fill. | Glaciogenic underlain*; PAZs; Lusi. biota.; OSL, AAR | 12 | 12 | 11 | 2 | 0 | 0 |
| Belgium *Scheldt estuary* | Never glaciated. Quaternary valleys and marine platforms cut into Paleogene clays and sands | PAZs, OSL | 8 | 1 | 1 | 1 | 0 | 0 |
| NW France *Calais* | Never glaciated. Cliffs, beaches, abrasion platforms, cut into Chalk and locally older substrate. | TL/OSL | | | | 1 | 0 | 0 |
| German Bight (GER) *Wadden Sea Elbe estuary* | Saalian (MIS-6) paraglacial*. Substrate is further Quaternary sedimentary basin fill. | Glaciogenic underlain*; PAZs | 6 | 2 | 5 | | | |
| SW Denmark *'Bakkeoer'* | Saalian (MIS-6) paraglacial*. Substrate is further Quaternary sedimentary basin fill. | Glaciogenic underlain*; PAZs; FAZs; AAR; Lusi. biota. | 6 | 2 | 3 | | | |
| N Denmark *Skagen, Anholt* | Last Interglacial equivalent of Skagerrak; Last-Glacial ice-overridden and GIA rebound affected. | Glaciogenic under and overlain; FAZs; Lusi. Biota, AAR | 0 | 6 | 0 | | | |





| Region | Description | Method | | | | | | |
|---|---|---|---|---|---|---|---|---|
| SW Baltic (GER) *Kiel* | Saalian (MI-6) paraglacial landscape, but Last Glacial ice overridden; ice-margin glacio-tectonized. Quaternary substrate. | Glaciogenic under & overlain; FAZs; PAZs; Lusi. biota. | 0 | 3 | 0 | | | |
| North England (UK) | Saalian (MI-6) paraglacial landscape, Last Glacial ice overridden / glacio-tectonized. Chalk substrate. | Glaciogenic under and overlain; | 4 | 0 | 1 | | | |
| East Anglia (UK) *Norfolk, Suffolk Ipswich* | Glaciated several cycles before Saalian (pre MIS 6). Cliffs, beaches, and transgressed valleys developed in till sheet and outwash overlying Chalk. | Biostratigraphic (macro- and micro-fossil), AAR | 1 | 2 | 1 | 1 | 0 | 2 |
| Thames Estuary (UK) *London, Mersea* | Terraced main river valley; never glaciated (at times proglacial, pre MIS 6). Alluvium overlying Paleogene clays, Chalk.. | Biostratigraphic (macro- and micro-fossil), AAR | 0 | 0 | 2 | | | |
| *English Channel* | | | | | | | | |
| Hampshire & West Sussex (UK) | Cliffs, bays, coves, raised beach flights; never glaciated. Cut into Chalk and older bedrock. | OSL, AAR | 3 | 6 | 5 | 0 | 5 | 4 |
| Offshore | Shelf floor lowstand valleys | OSL | 0 | 1 | 1 | | | |
| Devon (UK) | Cliffs, bays, coves, never glaciated. Abrasion platforms and valleys cut into Devonian sandstones. | Speleothem U-series | 1 | 0 | 0 | | | |
| Normandy (F) *Cotentin* | Cliffs, bays, coves, raised beach flights; never glaciated | Morphostratigraphy, OSL, TL | 10 | 1 | 3 | | | |
| Channel Islands *Jersey* | Cliffs, bays, coves, raised beach flights; never glaciated | Morphostratigraphy, U-series | 3 | 0 | 0 | | | |
| NE Brittany (F) | Cliffs, bays, coves, raised beach flights; never glaciated | OSL | 4 | 1 | 1 | | | |
| *Further West* | | | | | | | | |
| SW Wales (UK) | Cliffs, bays, coves, raised beach flights; near Last Glacial ice margin | AAR | 3 | 0 | 0 | | | |
| W Cornwall (UK) | Cliffs, bays, coves, raised beach flights; near Last Glacial ice margin | AAR | 3 | 0 | 1 | | | |
| W & S Brittany (F) | Cliffs, bays, coves, raised beach flights; never glaciated | Morphostratigraphy | 5 | 0 | 0 | | | |
| | | | | | | | | |
| TOTALS | | | 69 | 37 | 35 | | | |

\* paraglacial in terms of setting and glaciogenic underlain in terms of relative age control mean that (i) the Eemian transgression (in the first half of MIS 5e) affected landforms created by the Saalian glaciation-deglaciation episode (during MIS-6). And (ii) that one can commonly separate the Eemian interglacial deposits from older interglacial because they directly overly widespread traceable deposits produced during the preceding glaciation. This is the reason the area is considered the target area (i.e. type area) for correlative stratigraphic methods (e.g. Hansen, 1965, Miller and Mangerud, 1986, Zagwijn, 1996).



## 2 Literature overview

The dataset documented in this paper comes from several countries and physiographically diverse lengths of LIG coastline.
The research history on the subject is long, going back to the 19th century when marine molluscan biostratigraphic evidence
identified the buried Last-Interglacial equivalent of the North Sea (Harting, 1874; Lorié, 1906; Nordmann, 1928), and along
the south coast of England (Dixon, 1850; Reid, 1898). Last-Interglacial deposits have been studied intensively during the
20th century (e.g. Zagwijn, 1961; Müller, 1974; Menke and Tynne, 1984), which gradually established the 'Eemian' (though
widely termed the 'Ipswichian' in British regional literature, Table 1) as a superregional chronostratigraphic unit (e.g.
Zagwijn, 1996; Turner, 2000), more or less equivalent to the LIG and MIS 5e (e.g. Cohen and Gibbard, 2019). As such, its
onset, acme and terminus of the interglacial became a correlation target across Europe and the North Atlantic in terrestrial
climatological studies (e.g. Tzedakis, 2003; Sirocko et al., 2005; Helmens et al., 2015), as well as in glaciological (e.g.
NEEM community members, 2013), oceanographic (e.g. Bauch et al., 2011), and shallow and coastal marine research (e.g.,
Long et al., 2015).


For reasons of brevity and focus, we restrict the literature overview here to studies addressing LIG relative sea level (RSL)
positions. Note that there is also a wealth of LIG terrestrial sites in the UK and mainland Europe, many of which have been
used to understand regional climatic and environmental change during the LIG (Behre et al., 2005; Kühl et al., 2007; Candy
et al., 2016). In the database entry, we restricted ourselves to the lowest elevated terrestrial-only sites that serve in sea-level
reconstructions as 'terrestrial limiting' data points. Adding higher elevated sites, as close to modern and palaeocoastlines as
they may be spatially (e.g. Folkestone Battery at +27 m OD; Bridgland et al., 1995b), provides no added constraint to sea-
level reconstruction. Discussion of older interglacial only serves to separate and unmix them from LIG inventory, in the
present state of research.

### 2.1 North Sea: Netherlands onshore and offshore

A key paper and starting point for any literature overview on LIG sea-level indicator points in the southern North Sea, is that
by Zagwijn (1983) entitled *Sea-level changes in the Netherlands during the Eemian*. The paper draws comparison with
Holocene RSL reconstruction in the same region, but more so informs (i) the setting of the Eemian deposits amidst Saalian
glacial landforms (Table 2), and (ii) the relative age control within the interglacial as offered by regional pollen assemblages
('pollen association zones', PAZs) of know durations  (PAZs E1-E6; Table 1), as they became established by Zagwijn and
Danish, British and German counterparts between the 1920s to 1980s (e.g. Jessen and Milthers, 1928; Müller, 1974; Menke
and Tynne, 1984).  All Zagwijn's considered locations, are positioned north of the Saalian maximum glaciation limit:
transgressive and highstand sites either overly Saalian tills and outwash (North Sea sites, N Netherlands sites) or are from
along tongue-basin/push-moraine embayments (central Netherlands sites). The falling stage sites are from the Rhine estuary.



This river in the Late Saalian, Eemian and Early Weichselian was positioned north of the Saalian limit separate from the
Meuse (Fig. 1), unlike in the Holocene situation (Van der Meene and Zagwijn, 1978).

The paper's summary diagram (reproduced in Fig. 2a) is a curve connecting 8 data-bars on a schematic time axis, which
Zagwijn (1996) replotted as 9 data points on a linear time axis (durations established via varve counts at a site in NW
Germany by (Müller, 1974) with PAZ E1-E6 totalling some 11,000 years, but floating on an absolute timescale.   It
encompasses a period of rising RSL (>20 m within 1000 years), a slowdown and highstand (over ca. 6000 years) and a sea
level drop during the late interglacial and Early Weichselian. These data points have been included in many later global data
compilations (e.g. Kopp et al., 2009), but because the absolute age of the E1-E6 scheme is ill constrained and remains
debated (see section 6.1), different authors have used and placed the start of the RSL curve at different points ranging from
~130 ka to ~120 ka (discussed further in Long et al., 2015).

Zagwijn's RSL curve plots 'high-tide levels' and during PAZ E5 plateaus at -8 m MSL for the inland most site Amersfoort
(Fig. 1). In plotting the data and constructing the curve, no vertical corrections were applied. The inland highstand, however,
is from a more gradually subsiding part near the edge of the North Sea basin, whereas the seaward sites used to trace
transgressive and falling stage sea levels are from locations closer to and over the Pleistocene depocenter (Fig. 1). The latter
sites experienced greater rates of background (i.e. non-GIA) subsidence since depositions, for which one should correct
vertical positions (see also section 4.2.1) when the data is to be used in global assessments evaluating ice loading/unloading
history and sea-level response (WALIS database's application goal). Correction for vertical land motion (VLM) reduces the
earlier quoted value for the rate of sea-level rise during E3 from '>20 m/kyr' to '>12 m/kyr', bringing it more in line with
rates seen in the southern North Sea at the onset of the Holocene (Vink et al., 2007; Hijma and Cohen, 2019). Most
adoptions of the 1983 curve in the later literature tie the Zagwijn-1983 data points to individual locations and apply
differential vertical corrections (e.g. Lambeck et al., 2006; Kopp et al., 2009), for which basin subsidence values in Kooi et
al. (1998) are typically used. The datasets likewise has included VLM specification for the North Sea basin sites (Section 4).

In the late 1980s, offshore investigations with participation of geological surveys from all North Sea countries revealed the
location of the initial Eemian transgression well offshore the northern Netherlands (Sha et al., 1991; Beets et al., 2008; site
BH89/2). New datapoints also became available from the NE Netherlands, from geological survey mapping campaigns (Ter
Wee, 1979; Bosch, 1990). In 1990s the infill and setting of the Amsterdam and Amersfoort basins were subject of detailed
multi-proxy studies, aiming to resolve the course of events of the Saalian deglaciation and the establishment of the Eemian
interglacial optimum (Van Leeuwen et al., 2000; Cleveringa et al., 2000). Focus was on chronostratigraphy,
paleoclimatology and general depositional history more so than on improving reconstruction of past sea level. A cross-
section from the centre to the southeastern rim of the Amsterdam Basin (De Gans et al., 2000) confirmed the presence of
intertidal deposits from the Eemian highstand, echoing Zagwijn (1983) observations along the rim of the Amersfoort Basin.


At this time, amino acid racemisation (AAR) dating of marine mollusca shells was performed on newly collected and archived materials (Miller and Mangerud, 1986), confirming the Eemian age of the offshore and onshore mollusc bearing

beds in Zagwijn (1983). Older interglacial marine levels do occur below the Saalian glaciation contact and associated deposits, which produced older AAR results.

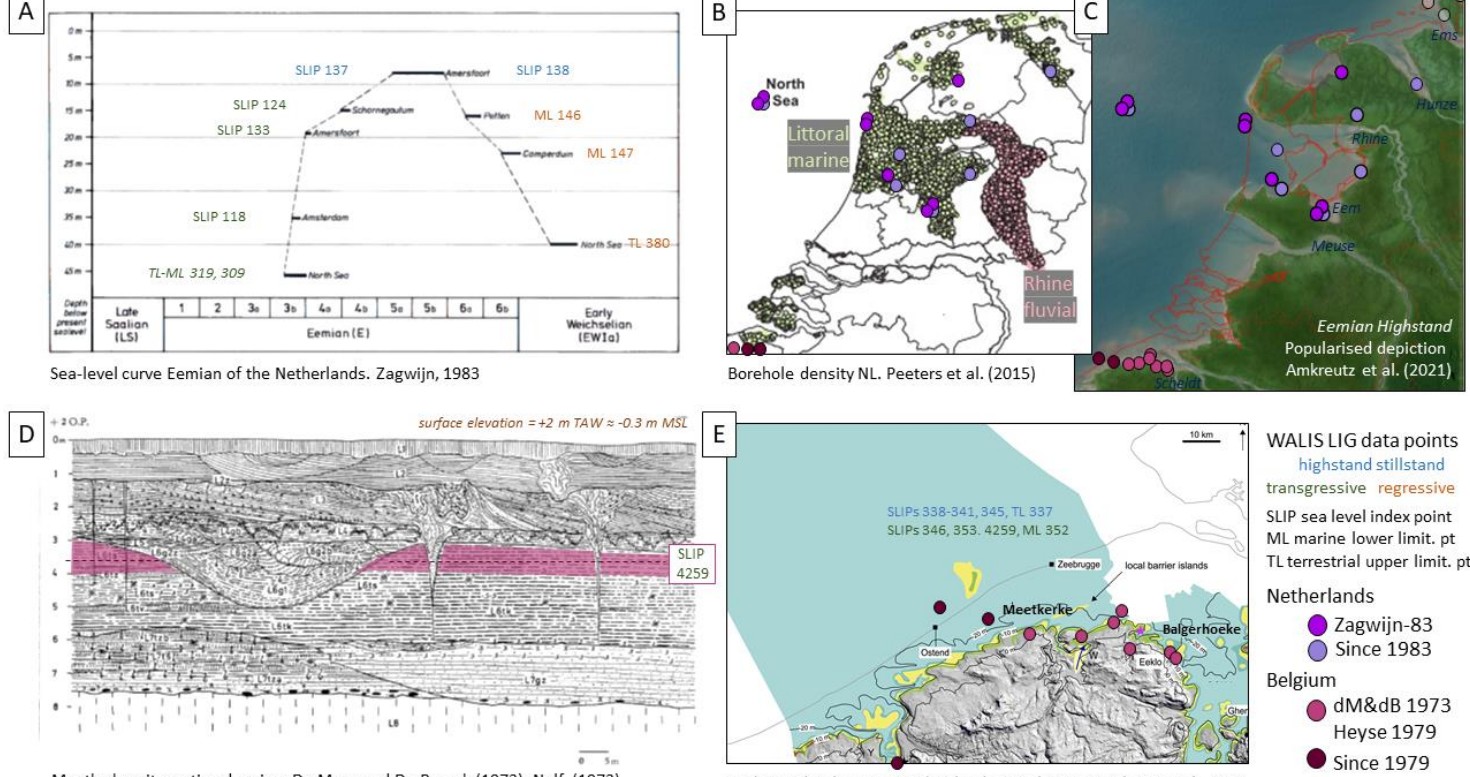

**Figure 2** Selected figures illustrating research history and database contents for The Netherlands (panels A-C) and Belgium (panels D-E), with annotations (database IDs, grouping in transgression, stillstand, highstand). A) sea level curve of Zagwijn (1983) B) Borehole density for the Netherlands (Peeters et al., 2015) overlain with legacy and more recent RSL data points. C) Popularised depiction of Eemian Highstand (Doggerland exhibition, Rijksmuseum voor Oudheden, Leiden; Amkreutz et

al., 2021). D) Exposure drawing of pit Meetkerke, Belgium (De Moor and De Breuck, 1973), elevation of SLIP highlighted. E) Paleogeographical map for Eemian highstand in Flanders, for a series in De Clercq et al. (2018). Panels reproduced with permission (A-E: Stichting NJG; Elsevier; RMO/Odé; present holder unknown, archived by Vliz.be; Wiley).

Since the late 1990s, investigation concentrated on the sedimentary development and chronostratigraphy of the river Rhine during the Late Saalian, Eemian and Weichselian (Table 1). Initially, this focused on areas south of the Saalian maximum ice limit (Törnqvist et al., 2000; Wallinga et al., 2004; Busschers et al., 2005, 2007; Hijma et al., 2012), where falling RSL erosion and reworking dominated and hence no Eemian sea-level constraints are preserved (Fig. 1; Table 2). The research





was important since the application of absolute optically stimulated luminescence (OSL) dating confirmed the correlation of
the Saalian maximum glacial limit to a phase within MIS-6, and the timing of the Eemian climatic optimum and North Sea
highstand in MIS 5e. The numeric age of the ice-limit landforms in the Netherlands and adjacent Germany was since
corroborated by OSL-ages at tens of sites (Busschers et al., 2008; Lang et al., 2018) clustering around 170-150 ka (Table 1:
Drenthe substage). Peak outwash sediment fluxes occurred at the same time in shelf-edge submarine deposits off the English
Channel (Armorican fan; Toucanne et al., 2010). Between the Saalian maximum limit and that of the Last Glacial (Fig. 1),
the glaciogenic terrain was formed between 160 and 130 ka mainly (Warthe submaximum at ~140 ka; Table 1)
Transgressive and highstand coastal deposits onlap Saalian-age glaciogenic landforms and river valleys, or they onlap the
floors and rims of valleys that dissected such landforms. OSL dates from within the Late Saalian and Eemian units confirm
their deposition occurred between 140-110 ka (Busschers et al., 2007; Peeters et al., 2016).

The river Rhine significant shifted course during the late Quaternary, from its northerly route occupied during the Saalian
deglaciation and throughout the Eemian, to the southerly course that it shares with the rivers Meuse and Scheldt . This shift
explains the absence of preserved Eemian coastal sites in the subsurface of today's Rhine-Meuse delta immediately south of
the Saalian glacial limit. In the last 10 years, the northerly Rhine course and the estuarine mouth that it developed during the
Eemian have received significant research attention (Sier et al., 2015, Peeters et al., 2016; 2019), yielding two new sea-level
sites at relative inland lagoon-rim positions: Rutten and Oosterwolde (Peeters et al., 2016). The classic Amersfoort locality
(Zagwijn, 1983), of which the data collection goes back to Lorié (1906) and Zagwijn (1961), was re-cored in 1980/81 (Miller
and Mangerud, 1986), in the 1990s (Cleveringa et al., 2000; two boreholes) and again in 2008-2015 (TNO Geological
Survey of the Netherlands; Durham University, Utrecht University), but has not yet produced new sea-level indicators.

## 2.2  North Sea: Belgium, Southern Bight

LIG coastal intertidal deposits in Belgium are encountered at shallow depths along the inland rim of the modern coastal
plain, typically separated from Holocene equivalents by periglacial deposits of Last Glacial age. From the very NW of
Belgium, the monograph of Heyse (1979) provides a series of pit exposures, describing the sedimentology and palynology
(using the E1-E6 scheme, as in The Netherlands). Further LIG deposits from estuarine environments are associated with
valley fills underlying the Flanders coastal plain, the main one cut by the river Scheldt in Middle Pleistocene times
(Flemmish Valley; Gibbard, 1988; De Moor and Pissart, 1992), and in the SW a second one of same broad age cut by the
river IJzer (Bogemans et al., 2016; hosting deposits of older interglacials too). De Moor and De Breuck (1973) and Nolf
(1973) describe the tidal sedimentology resp. marine molluscan content of pit site Meetkerke (Fig. 2d), just west of Brugge.
Authors Mathys (2009) and De Clercq et al., (2019) have mapped the onshore-offshore continuation of these valley systems,
the former with morphostratigraphic attention (intersection of Last Glacial dissective phenomena of the Dover Strait;
Bridgland and D'Olier, 1995; Gupta et al., 2007), the latter with particular attention to palaeogeography in successive stages
of sea-level rise and fall during the LIG (Fig. 2e). Compared to the fairly straight present-day coastline, the Eemian coast



line is considered a more irregular one, with Paleogene clay hilly outcrops forming capes and islands. The cluster of sites documented by Heyse (1979), occupies a position just north of such an outcrop. River valleys functioned in the valleys before and after the LIG. As in most lower reach valley settings, the falling and lowstand stage induced considerable erosion

to LIG deposits of the Belgian valley fills. Onshore, highstand deposits were left preserved just locally along valley rims. In the vicinity of the modern coastline, units from early transgressive stages did preserve locally below the falling stage erosive contacts. During youngest falling and low stand stages (Table 1), the Scheldt river established northwestward to northward valleys into the SW Netherlands (Vandenberghe, 1985), further explaining the degree of preservation of LIG deposits in the NW of Belgium and spatial distribution of WALIS entries for Belgium.

## 2.3 North Sea: German Bight, NW Germany, SW Denmark

LIG coastal deposition in and along the German Bight accumulated and preserved in very similar setting, as those of the central and Northern Netherlands. The research history also starts similarly early as in England and the Low Countries, with boreholes at Tønder (Madsen et al., 1908) and Højer (Nordman, 1928) in the Vidå valley particular classic sites. The area lay outside Scandinavian ice sheet coverage of the Last Glacial, though outwash produced along that margin was a major

eroding and burying agent to the Eemian coastal record. The substrate for the Eemian transgression is hosted by landforms generated during the Saalian glaciation, deglaciation, readvance (Drenthe-2 and Warthe morainic lines) and (glacio-)fluvial dissection (conduits of Vidå, Eider, Elbe, Weser and Ems). Miller and Mangerud (1986) resampled sites from along the Danish Eemian North Sea. Streif (2004) summarised evidence around the German Bight, linking up to Dutch, Belgian and English North Sea coastal sectors, and providing intercomparison with older interglacials and the Holocene. Konradi et al.

(2005), in their paper entitled *Marine Eemian in the Danish eastern North Sea*, provide palaeogeographical details for an inventory of sea-level indicator sites. They aptly characterise the Danish Eemian shoreline as 'more irregular than present' (Fig. 1). Some of the sea-level data points are reported from environments resembling those of the modern Wadden Sea, whereas others come from what would better be called 'fjords' and 'straits' cut in the Quaternary glacial and riverlain substrate of the freshly exposed glacial landscape at the onset of Eemian transgression. A palaeogeographical particularity is

a W-E marine highstand passage between German Bight and SW Baltic Sea across the German-Danish border (Schleswig-Holstein), unlike the peninsular present situation (Winn and Erlenkeuser, 1995; Konradi et al., 2005; see section 2.4). Such a connection is suspected from geological mapping the dented inland extent of the German Bight Eemian highstand (e.g. at Schnittlohe, ID 889, N Germany; Menke 1985; Knudsen 1985) and from biotic changes in the SW Baltic halfway the regional extensive Cyprina Clay marine beds (Funder et al., 2002; see section 2.4), although the supposed connection would

have been shallow and is not fully traced and hence not evident (Miller and Mangerud, 1986; Kosack and Lange, 1985; Schulz et al., 2001 - op cit. Konradi et al., 2005).

Continued sea-level rise towards the Eemian highstand inundated good parts of the German Bight's Saalian glacial derived push-moraine and till-plateau islands (dubbed 'Bakkeør'), which were subsequently hidden by LGM and deglacial sediments





and basin-tectonic and GIA subsidence. Between the Bakkeør-island series and the Danish mainland a retreat-stage meltwater plain is present, which steered the transgression in the Danish sector and drew considerable investigation (listed in Konradi et al., 2005:24). Southwest of the Bakkeø-islands/Warthe limit, lay the Elbe valley, an ice-marginal river during the Warthe ice-limit and the LGM, which formed an Eemian estuary preserving LIG sea-level indicators recorded by Menke and Tynni (1984); Menke (1985) and Streif (1990; 1991). West of the Elbe valley/estuary, a secondary morainic limit exists (Fig.

1; e.g. Ehlers et al., 2004), and the associated lows host intensively studied Eemian and Early Weichselian terrestrial deposits (Selle and Schneekloth 1965; Behre et al., 2005; key sites Oerel and Glinde). The northernmost, deepest sites also preserve evidence of marine incursions (Streif 1990, 2004). Further West, the till plateau of Ost-Friesland and minor valleys set into it host Eemian coastal deposits, similar to sites further west along the Wadden Sea in the Netherlands.

Largely age-control of the NW German and SW Danish sites (Table 2) is based upon pollen investigations on terrestrial beds immediately below and above tidal coastal deposits and contained in the brackish and saline deposits, resolving broad-scale vegetation succession (i.e. Zagwijn E1-E6 scheme; converted to from Danish and German origin schemes; e.g. Funder et al., 2002). Like in the Dutch sector, the Saalian deglaciated geological-geomorphological setting means identifying LIG from older interglacial deposits is relatively simple. Importantly, no inland fluvial-estuarine sites are known for the German Bight:

falling RSL and lowstand erosional activity during the last glacial cycle (Table 1) appears to have mostly removed expected Eemian highstand morphostratigraphic records from along the valleys of Elbe, Weser and Ems. This is in contrast to the Rhine, Scheldt and Thames estuaries in the Dutch, Belgian and English LIG North Sea coastal segments.

**2.4 Skagerrak and SW Baltic: N Denmark, NE Germany – within the Last Glacial limit**

North and Eastward into Denmark, land and seafloors were ice covered during the Last Glacial, meaning that LIG coastal

deposits were eroded and glaciotectonically displaced. In NE Denmark, deep boreholes on the Jutland peninsula and islands facing the Skagerrak-Kattegat embayment reveal thick marine sequences overlying Saalian-aged till and below Last Glacial erosive contacts, exclusively from relatively deep marine environments (Jessen et al., 1910; Bahnson et al., 1974; Knudsen and Lykke-Andersen 1982; Knudsen et al., 2009). These sequences record regional details of the oceanographic-climatic amelioration (warming, invading boreo-Lusitanian biota) at the onset of the LIG, as well as deterioration (cooling, returning

boreal and subpolar biota) following the LIG, e.g. at Anholt (Seidenkrantz, 1993: Kattegat stadial just prior to beginning of Eemian), and Skaerumhede, North Jutland (Miller and Mangerud, 1986; Houmark-Nielsen, 1987; Larsen et al., 2009). At these locations, the Skagerrak-Kattegat seabed appears to have been overridden by Last Glacial ice, but only superficially eroded, and lower parts not laterally displaced. Similar to Holocene RSL evidence from the same region (raised beaches of Middle Holocene age at +12 m at Skagen Odde; Clemmensen et al., 2001; 2018; Rosentau et al., 2021), considerable GIA

vertical movement will have affected these north-easternmost sites in our compilation. Note that a separate regional LIG sea-level database paper will cover Scandinavian, Polish, Finnish, Baltic and Russian regions (Dalton et al., 2021/this special issue).

In NW Denmark coastal marine Eemian deposits occur at shallower depths, not dissimilar to these in areas unaffected by
glaciation in SW Denmark. Their positions within the Last Glacial limit and the level of disclosure that borehole-based
subsurface mapping can give, however, means that some displacement cannot be excluded. Sites in this part of Denmark
provide information on palaeocoastline positions (Fig. 1), but less so for reconstructing sea-level elevations (Konradi et al.,
2005).

In SE Denmark, modern day cliff sections in glacial deposits on the isles facing the Baltic, such as Ristige Klint and Aero
(Madsen et al., 1908), provide outcrops of shallow marine, coastal and near-coastal terrestrial beds, which have been
extensively studied in a paleoenvironmental content (overview in Kristensen et al., 2000, Funder et al., 2002), and OSL-
dated (Murray and Funder, 2003; Buylaert et al., 2011). These sites allowed for fairly-detailed palaeoceanographic
reconstruction for the Baltic Sea during the LIG, in particular what connections it maintained and established with the North
Sea. The LIG marine deposits in the SW Baltic are known as the Cyprina Clay, which biota reveal to have evolved through a
short initial brackish phase, and a longer twofold saline phase. The clay is heavily displaced and exposed in SE Danish
coastal cliffs (Madsen et al., 1908; Kristensen et al., 2000), and stretches southwest and southeast into Germany with
incursions near Lubeck, and outcrops at Rügen and Usedom (Madsen et al., 1908; Konradi, 1976; Houmark-Nielsen, 1987;
Kubisch and Schoenfeld, 1985; Ruehberg et al., 1995; Winn et al., 2000).


For the SW Baltic Sea, mollusca and foraminifera associations at the base of the Cyprina-Clay, and geological mapping of
N-S running straits separating the Danish Isles connecting the Baltic to the Kattegat (similar to the geography today), reveal
transgressive marine connections along these N-S routes (Kristensen et al., 2000; Funder et al., 2002). As the interglacial
unfolded, the salinity and temperature signals in the biota markedly change further (Kristensen et al., 2000) and, importantly
for sea-level reconstruction, the connection across Schleswig-Holstein may have established (Funder et al., 2002). Pollen
palynological investigations from the Cyprina Clay allow correlation of the SW Baltic oceanographic phases to the terrestrial
forest succession schemes (Table 1), the brackish stage to PAZ E2b/E3a and division of the saline phase during PAZ E4b
(Kristensen et al., 2000), consistently to the NW Danish marine records (Seidenkrantz 1993, Larsen et al., 2009). The Baltic
Sea basin during the LIG has also connected to the White Sea over nowadays NW Russian (Ladoga lake; Karelia), the
timing of which is independent (Funder et al., 2002) from the events and foraminiferal, boreal and Lusitanian mollusc
evidence connecting the Cyprina Clay to North Sea incursions.

Regression along the SW Baltic is not well vertically resolved, as would-be littoral sequence there experienced subsequent
subglacial erosion by the Scandinavian ice-sheet expanding across it during the Last Glacial, and deglacial fluvial erosion
and reworking (Meng et al., 2021). In the deeper water paleogeographic setting on the Danish side, the Baltic is marine well
into the Early Weichselian, owing to its deep N-S connections to Kattegat and Skagerrak. The secondary connection W-E





across Schleswig (Fig. 1) is considered to have fallen dry early on in PAZ E6, as site Schnittlohe then shows a return to limnic conditions (section 2.3). Limnic and peatland deposits replaced littoral environments at the end of the interglacial (PAZ E5/6 transition) also in Baltic Sea palaeobays in NE Germany (Meng et al., 2021), but the evidence for this is from

displaced and reworked contexts and no sites have provided opportunity of water level reconstructions. East of the area covered in this paper, terrestrial deposits topping estuarine ones in the Vistula (*Wisla, Weichsel*) valley sequence give similar indication (site Nowiny, Poland; Makowska 1986 op cit. Funder et al., 2002; Drozowksi, 1988 op cit. Lambeck et al., 2006).

## 2.5  North Sea: Thames Estuary, East Anglia, North England

The stratigraphy of the Thames region of eastern England has been extensively studied and documented from the 19[th]

century as the development of London and the surrounding area exposed multiple phases of cold-riverine and warm organic interglacial sediments, many with marine and freshwater micro and macro-fossils.  The discovery of Eemian *Hippopotamus* remains during excavations of Trafalgar Square in the 1950's (Franks, 1960) garners continued wider attention; but numerous less publicised investigations have identified previous interglacial brackish, coastal and shallow marine sediments (e.g., Abbott, 1892; Gibbard, 1985; Preece, 1999; Sparks and West, 1963; Hinton and Kennard, 1901).  Similarly in East

Anglia, local site investigations document the presence of brackish or marine molluscs at several sites corresponding to the 'Ipswichian' (Eemian) interglacial based upon pollen stratigraphy, for example at March (Whitaker et al., 1893) and Wretton (Sparks and West, 1970).  Further north, evidence of LIG sea level is sparser with much of it removed by the subsequent Last Glacial (Devensian) ice advance along the coast of North East England and Yorkshire (Sutherland et al., 2020; Catt, 2007), and is largely restricted to the Sewerby raised beach, East Yorkshire (Bateman and Catt, 1996; Lamplugh, 1887; Catt

and Penny, 1966), which is now buried under a landslide.

A major challenge is that interglacial sites from England, unlike in the Dutch-German-Danish contexts where Saalian glaciogenic landforms and substrates allow to separate LIG from older interglacial sites, often appear in ambiguous stratigraphic positions. Whereas palynologically and using molluscan assemblages they can be identified to be from warm

periods and near-coastal environments, the sites lack independent dating, and pollen sub-stages of the 'Ipswichian' and the preceding 'Hoxnian' interglacials have very similar vegetation profiles (Thomas, 2001; Turner and West, 1968).  Attempts to correlate seven, then termed, 'Ipswichian' estuarine sequences from the Thames, alongside those from East Anglia, based upon pollen sub-stages and their relative elevations, resulted in a paper by Hollin (1977) on *Ipswichian sea levels and Antarctic ice surges* identifying the potential for a double LIG sea-level highstand, investigating the idea of a late, rapid

collapse of the Antarctic ice sheet.  Though in keeping with more recent hypotheses (e.g., O'Leary et al., 2013), significant advances in AAR methods (in particular dating freshwater *Bithynia tentaculata* opercula (Penkman et al., 2013)), combined with Thames terrace stratigraphy (Bridgland, 1994), vertebrate and molluscan biostratigraphy (Schreve, 2001; e.g., Preece, 2001) and independent geochronology (e.g., OSL) has since shown that many British 'Ipswichian' sites actually date from the preceding three marine isotope stages (Penkman et al., 2011; 2013; Penkman et al., 2008).  Trafalgar Square is the only



site in Hollin's (1977) analysis which is still regarded of LIG age.  Ilford, Aveley and Crayford are now known to have been deposited during MIS 7; Purfleet and West Thurrock during MIS 9; and Little Thurrock in MIS 11 (Penkman et al., 2011; Penkman et al., 2008). Likewise the Speeton Shell Bed in East Yorkshire (Lamplugh, 1881; West, 1969) is now thought to date to MIS 7 (Wilson, 1991).

### 2.6  English Channel: Southern England

LIG deposits are found along the length of the southern coastline of Britain, typically as part of expansive sequences of raised littoral terraces from the Pleistocene. Occurrences of Pleistocene deposits that formed during interglacials are well documented across all southern coastal counties (Prestwich 1892) in early literature; from Kent in the south east (Mackie 1851), through Sussex (Godwin-Austen 1857; Prestwich, 1859; Reid 1892), Hampshire (Reid 1893) and Dorset (Weston 1852; Prestwich 1875), to Devon and Cornwall (De La Beche 1839, Ussher 1879) in the southwest. Pleistocene littoral

deposits occur around Land's End (the southernmost point of mainland Britain) and can be found on the northern coastlines of Cornwall and Devon, the east coast of Somerset (Woodward 1876) and from there across the southern coastline of South Wales (Murchison 1868; Prestwich 1892). Unsurprisingly, the early literature associated with these widespread deposits extends to include raised littoral deposits on the main island groups in the English Channel: the Isle of Wight (Godwin-Austen 1855, Prestwich 1859), the Isles of Scilly (Barrow 1906), and the Channel Islands (Zeuner 1946).


Early literature typically relied on the identification of mammalian and molluscan fauna, as well as pollen assemblages and successions, to attribute deposits to successive interglacials. The elevations of littoral terraces (typically, raised beach deposits) and apparent positions in flights of these were also used. The relative elevations of deposits also became a useful tool for tracking contemporaneous raised littoral terraces across wider geographical areas (Palmer and Cooke 1923; Arkell

1943; Mitchell 1960; Orme 1960; West and Sparks 1960; Hodgson 1964). A commonality that emerged was that the Ipswichian (Eemian/MIS 5e) interglacial was likely associated with the lowest terrace of deposits, often reported as a '15 foot raised beach' (c. 4.5 m O.D.) (Fig. 3a). However, association with the preceding interglacial (MIS 7) was hard to rule out, especially given the presence of multiple, distinguishable low-level raised beach deposits at several locations (Davies and Keen 1985; Mottershead et al., 1987; Bates et al., 1997). Dating attempts often remained inconclusive and debated

(Mitchell 1972; Bowen 1973). Unfortunately, despite the development and application of novel quantitative dating approaches during and since the 1980s, many low-level raised littoral deposits in southern and southwest Britain still lack confident and precise age determinations (see further discussion in section 6).

**Figure 3 Selected figures illustrating research history and database contents for South England (panel A) and Northwest France (panel B), with annotations (database IDs, grouping in transgression, stillstand, highstand). A) Schematic section of raised beach series (Bates et al., 2010), with SLIP producing and ML producing sites annotated. B) Wave-cut platforms (PFs) as mapped for NE Cotentin (Coutard et al., 2006), annotated with LIG SLIPs, MLs and TL locations, associated to the PF I level. Panels reproduced with permission (A, B: Elsevier).**



**2.7 English Channel: Northwest France**

The northwest coast of France, from Britany to the Straits of Dover, is characterised by the occurrence of a staircase of polygenic coastal platforms that were correlated with Pleistocene interglacial periods. These platforms are intermittently covered by a variety of coastal (mainly raised beaches) and periglacial deposits (heads and loess) that have been studied for more than a century (Barrois, 1877, 1882; Bigot, 1885, 1930; Guilcher, 1948; Pellerin and Dupeuble, 1979; Lautridou et al., 1999; Regnauld et al., 2003; Coutard et al., 2005, 2006; Cliquet et al., 2009; Pedoja et al., 2018). The absence of well-
preserved fossiliferous remains in most of these sites gave rise to an extensive discussion about their chronostratigraphy, and the age was generally established based on the altitude of the beach deposits relative to other sites. In the Armorican Massif, two levels of marine deposits, between 0-5 and 15-20 m above sea level were attributed to the LIG (Elhaï, 1963), while Morzadec-Kerfourn and Monnier (1982) attributed the three lower levels, found between 0-15 m in Britany, to three distinct marine transgressions, probably MIS 9 to MIS 5.


The discovery and excavation of Palaeolithic sites along the coast and the progressive application of thermoluminescence (TL), optically stimulated luminescence (OSL) and electro spin resonance (ESR) geochronology (e.g., Balescu et al., 1991; Balescu and Lamothe, 1992; Loyer et al., 1995; Folz, 2000; Coutard et al, 2006; Cliquet et al., 2003, 2009; Monier et al., 2011) boosted the understanding of the coastal sequences and helped to better constraint the distribution of the LIG deposits.
Loyer et al., (1995) suggested that up to four different interglacials were preserved in the same coastal platform in Saint-Brieuc Bay (Britany) based on palaeoenvironmental data and TL dating of loess deposits, later further supported by ESR dating of Palaeolithic sites (Bahain et al., 2012). Thermoluminescence dating of burnt flints in Port-Racine (Normandy), attribute beach deposits at 3 m to the Eemian (Cliquet, 1992, Lautridou et al., 1999), and Coutard et al., (2006) OSL dated several marine, beach and dune deposits in Val de Saire (Normandy; Fig. 3b) pointing to the MIS 5e to 5d. Despite the
fragmentary nature of the deposits and the relatively small number of absolute age determinations, there were also some attempts to reconstruct the LIG coastline in Normandy (Regnauld et al., 2003) and to establish regional stratigraphic correlations across the English Channel (van Vliet-Lanoë et al., 2000; Bates et al., 2003). Deposits corresponding to the LIG appear better preserved in Britany and western Normandy (Fig 3b) where the local geomorphology protected these sediments from later erosion; however, further east Pleistocene beach deposits were eroded in younger interglacials and the Holocene
(van Vliet-Lanoë et al., 2000; Bates et al., 2003; Regnauld et al., 2003). As an exception, at Sangatte in NW France a cliff foot beach (De Heinzelin, 1966) that predates the LIG (Balescu and Haesaerts, 1984; Haesaerts and Dupuis, 1986; Balescu et al., 1991) is preserved under colluvium, re-exposed by Holocene erosion.



## 3   Sea level indicators

### 3.1 Regional context differences

On identifying and describing LIG depositional settings and interpreting relevance for sea-level reconstructions, comparison is often made to the modern setting. Figure 1 is an overview map depicting the approximate LIG coastline relative to current topography, while Table 2 lists the coastal settings region-by-region. Considerable spatial variation exist is the current coastline around the North Sea and English Channel, as did during the LIG highstand. Extensive lowland coastal plains with back-barrier tidal environments, marshes and lagoons dominate the NW Belgian-Dutch-German-SW Danish stretch of North

Sea coast, interrupted by headlands of Saalian age ice-marginal morphology (push moraine-outwash complexes, till sheet plateaus) and merging with estuaries and delta plains of main rivers (Elbe, Weser, Ems, Rhine, Meuse, Scheldt). The English side of the North Sea coast has cliffs in Middle Pleistocene till accumulations over Chalk bedrock at shallow depth along Lincolnshire and East Anglia, a series of smaller estuaries interrupted by Paleogene outcrops in Suffolk, opening into the Lower Thames estuary which reaches inland to central London.  Chalk cliff interrupted by estuaries such as the Medway

form the coast of Kent and the Strait of Dover. This marine reach connecting North Sea and English Channel is not a regular shelf area, but a transgressed, eroded gorge created by lowstand periglacial and proglacial outwash rivers (repeatedly since the Middle Pleistocene) (Gibbard, 1995; Bridgland and D'Olier, 1995; Gupta et al., 2007; Mellett et al., 2013). The British and French coasts on either side of the English Channel alternate inset estuaries (larger ones named in Fig. 1) with cliff and bluff sections (with gravelly to sandy beaches at their feet), in a variety of substrates (Mesozoic, Palaeozoic, crystalline).

Bedrock island features such as the Isle of Wight (British side), Channel Islands (French side) and Belle Isle (SW Brittany) existed in LIG and Holocene times alike.

When comparison is sought between the current and previous highstand, the geomorphological age and longevity of landforms that constrain the coastline must be considered, especially in the North Sea region with its intermittent glaciations

(Cohen et al., 2014; Cohen, 2017). Whereas the broad position of estuaries and cliff-beach settings along the English Channel and in the SW of the North Sea is essentially the same between the LIG and Holocene, the configuration of lagoons, river mouths and morainic islands in the Netherlands and the German Bight sectors of the North Sea (the area between Saalian maximum and Last Glacial limits in Figure 1) is fairly different. In this eastern North Sea region, the Eemian transgression and highstand affected a paraglacial landscape left after the Saalian deglaciation (Table 2). Furthermore, the

long-term basin subsidence (depocenter indicated in Fig. 1) have made Eemian paleo-coastal depositional architectures preserve as subcrop: highstand sea-level encountered at several meters below the equivalent modern sea-level position. This affects the way deposits are studied (e.g., boreholes v. outcrops) and the areal extent over which deposits tend to preserved (extensive basin-central patches, v. small pockets at the rims of a system). By comparison, western North Sea LIG features along the coast of the UK and France are preserved at higher elevations due to the relative lack of long-term subsidence.




## 3.2 Overview of Indicator Types

Table 3 presents an overview of the indicator types associated to the SLIPs, including regional statistics. In the WALIS database structure, terrestrial and marine limiting data entries are not further specified to type. Formal descriptions are provided for each indicator type. All indicator types have Holocene analogue environments in the same region. Most indicator types constrain RSL elevations by combining sedimentary properties, fossil content, and architectural evidence. In all cases, marine biota supported choices of specific Indicator Types. We used (in part: defined) 10 types of RSL indicator in this region (Table 3), and document the non-generic ones below.

**Table 3: Types of RSL indicators used in NW Europe database.**

| Name of RSL indicator | Indicator reference(s) | ID in WALIS | Region and # of occurrences |
|---|---|---|---|
| WALIS generic, *not further documented here – see other papers ESSD special issue* | | | |
| Marine Terrace | Pirazzoli, 2005<br>Pedoja et al., 2011<br>Rovere et al., 2016 | 7 | France (12 LIG, 3 older) |
| Beach deposit or beach rock | Mauz et al., 2015<br>Rovere et al., 2016 | 11 | France (7 LIG, 5 older)<br>United Kingdom (4 LIG, 9 older) |
| Beach ridge;<br>Beach swash deposit | Otvos, 2000<br>Rovere et al., 2016 | 12,<br>29 | France (1)<br>United Kingdom (2 LIG, 1 older) |
| Lagoonal deposit | Rovere et al., 2016<br>Zecchin et al., 2004 | 15 | Netherlands (3 LIG, 1 older) |
| Paper Author added as part of this study, *documented by sections 3.2.1 to 3.2.5.* | | | |
| Drowned valley floor Transgressive Contact | Vis et al., 2015<br>Peeters et al., 2016, 2019 | 17 | Netherlands (1)<br>Denmark (1) |
| Basal Peat<br>(non-mangrove) | Jelgersma, 1960<br>Van de Plassche, 1982<br>Zagwijn, 1983<br>Hijma and Cohen, 2019 | 18 | Netherlands (3)<br>Denmark (2) |
| Isolation Basin (moment of marine connection) | Zagwijn, 1983<br>Van Leeuwen et al., 2000<br>Beets et al., 2006 | 19 | Netherlands (3) |





| | | | |
|---|---|---|---|
| Estuarine Terrace (preserved tidal flat surface) | De Moor and De Breuck, 1973 Zagwijn, 1983 Peeters et al., 2016, 2019 | 20 | Belgium (9 LIG, 1 older) Germany (3) Netherlands (2 LIG, 1 older) |
| Salt marsh (various subtypes) | Englehart and Horton, 2012 | 37 | United Kingdom (2) |
| WALIS generic, *but region specific information provided in section 3.2.6* | | | |
| Shallow or intertidal marine fauna | Subregion specific Refs in section 3.10 | 33 | Germany (3) Denmark (3) United Kingdom (3 LIG, 2 older) |

### 3.2.1 Indicator type: Drowned Valley Floor

This indicator type relates to a contact between terrestrial depositional facies (below) and subaquatic depositional facies (above) and provides transgressive context SLIPs. The terrestrial facies is typically a decimetre thick river organic mud with immature palaeosol, if not a flood-basin peat bed, with further fluvial facies below [late lowstand palaeosols, evidence of climatic amelioration before transgression]. The subaquatic facies is typically a decimetre-metre-thick organic mud, rich in fine and coarse detrital organic matter, rich in silt admixture, bearing tidal indicators, bearing microfossil indicators of occasional brackish estuarine in wash. It grades upward into established tidal, brackish to saline, full estuarine facies. See Hijma and Cohen (2011, 2019) for Holocene examples and Sier et al. (2015) and Peeters et al. (2016, 2019) for LIG examples.

The transgressive contact has direct meaning as to estuarine inundation, and represents the SLIP. The secondary contact within the terrestrial facies can be used as a terrestrial limiting point. In wide valleys experiencing relatively rapid postglacial transgression, unfilled estuaries result in which tidal amplification (owing to estuary funnelling) is not yet a major factor (HAT values may in fact be dampened inland in such estuaries). Herein the estuary in a freshly drowned lowland valley differs from later stages of estuary development as observed in highstand filled estuaries (inland propagation, amplification and dissipation does affect estuarine type SLIPs). On the other hand, in inland parts of the estuary riverine discharge may impose a gradient and lift dampened inland tides to the same altitudes as that of HAT at the estuary mouth (Van de Plassche, 1995; Vis et al., 2015). For these reasons, the 'Drowned Valley Floor' base-estuarine indicator type is kept separate from the 'Basal Peat' and 'Estuary terrace' indicator types that are introduced below.

In formula (HAT = Highest Astronomical Tide; MSL is mean sea-level, a.k.a. half tide):

WALIS Relative Water Level description: (HAT to MSL) / 2

WALIS Indicative Range description:   HAT to MSL

### 3.2.2 Indicator type: Basal Peats

Basal Peats are terrestrial deposits encountered along the base of transgressive-to-highstand depositional systems and in submerged position on inner shelves (e.g. Jelgersma, 1979; Hanebuth et al., 2000). Using basal peats as RSL indicators became widely established in Holocene sea-level communities since the 1970s (Van de Plassche, 1986), both onshore and offshore. Jelgersma (1961) and Van de Plassche (1982, 1995) provide classic Holocene reference examples for The Netherlands. Likewise, it is useful as an RSL indicator in interglacial coastal settings (e.g. Zagwijn, 1983; Streif, 1990, 2004; Konradi et al., 2005), especially when combined with palynological investigations to provide time-control on the position within the interglacial (see section 3.3).

Basal peats are found underneath transgressive surfaces, buried by subaqueous deposits. Basal peats occur across valley floors, across valley rims, across interfluve highs, across flanks of isolated topographic features within older coastal marsh and subaquatic aggradational facies (above). The peaty terrestrial facies is typically a few decimeters thick (in compacted state, cf. Greensmith and Tucker, 1986; Brain, 2015), and often overlies a surface with a more developed paleosol. The latter are 'late lowstand' boreal to temperate paleosols, which indicate that notable climatic amelioration occurred before actual transgression, i.e. a time gap between climatic onset of interglacial and establishment of the highstand.

The very top of a Basal Peat bed indicates submergence of a swamp/marsh and may be recorded as a SLIP. The very base of Basal Peat bed indicates a palaeo-groundwater level (GWL) that in sea-level reconstruction context becomes a terrestrial limiting point. In argued cases such a limiting point may be upgraded to a SLIP, if it formed along the inland rim of a transgressive lagoonal or lagoonal-deltaic environment (e.g. Van de Plassche, 1986; Nelson, 2015). Assessing the palaeogeographical situation of the basal peat data point, or swarm of data points, may help to screen limiting points and upgrade the status of coastally most-relevant ones (Vis et al., 2015; Hijma and Cohen, 2019).

In formula:

    WALIS Relative Water Level description:   GWL

    WALIS Indicative Range description:       Swamp peat: (GWL to GWL-0.2m); Marsh peat (GWL-0.3 to GWL-0.8m).

### 3.2.3 Indicator type: Isolation Basin (marine connection moment)

Isolation basins are used extensively in Holocene sea-level studies in higher latitude coastal environments, making use of the prominent occurrence of lakes in freshly deglaciated environments, and the ecological sensitivity of such water bodies when connecting and disconnecting from the sea (e.g., Long et al., 2011; Shennan et al., 2005; Sundelin, 1917). In the North Sea area, substantial lakes formed when ice sheet cover from the penultimate glaciation disintegrated (during MIS 6). The period at which these lakes connected to the North Sea and transformed into highstand marine embayments, has been a primary constraint on the relative timing of the Eemian Transgression (e.g. Zagwijn, 1983). The reconstructed elevation of lake sills



(the lowest point of the basin rim) provides the elevation of SLIP of this type, whereas the contact between lacustrine environment (lower) and brackish-marine environment (upper) established in central part of a basin is the location where age control is obtained. Ideally the sill of an isolation basin is formed of unmodified bedrock (Long et al., 2011), whereas the lake shorelines in the North Sea are formed of glacial diamicton and/or glaciotectonised ridges (Zagwijn, 1983; De Gans et al., 2000), which means additional uncertainty as to their elevation must be given. Our WALIS database entries have

registered the latter location as coordinates. The site description mentions separately where the paired sill level is positioned. Relative Water level description and IR link to the sill location. Water depth of the lake is irrelevant for the application.

In formula:

WALIS Relative Water Level description:   (HAT to MSL)/2

WALIS Indicative Range description:        (HAT to MSL) + uncertainty sill level position

### 3.2.4 Indicator type: Estuarine Terrace

This indicator type is introduced as a variant of the Marine Terrace (cf. Pirazzoli et al., 2005) as used elsewhere in the WALIS database. Paraphrasing that description, it considers "any relatively flat surface of estuarine origin". A difference with the Marine Terraces is that the orientation of the estuarine terrace is 'along' a falling-stage river valley / highstand embayment, and 'across' the shoreline of headland coasts and barrier-coastal stretches.  Furthermore, the 'flatness' of the

abandoned surface is not so much due to wave action and storm swash, but more due to intertidal/supratidal flooding just prior to terrace abandonment. Estuarine terraces provide indicators for highstand and regressive contexts mainly. In landward direction, estuarine terraces grade to riverine terraces / former floodplains that provide terrestrial limiting point rather than SLIPs. The Estuarine Terrace indicator type is introduced next to Marine Terraces, to allow to assign different indicative meaning to elevations sampled from estuarine terraces where the 'flat' surfaces are usually formed in facies

bearing intertidal sedimentological indications (alluvial terraces), than to marine terraces that are flattened due to abrasion processes (straths). Site Balgerhoeke (Heyse, 1979; IDs 339, 340; Scheldt) and site Petten (Zagwijn, 1983; ID 146; Rhine) provide for highstand and regressive SLIP examples (see Figs. 1 and 2).

In formula:

WALIS Relative Water Level description: (HAT + MSL) / 2

WALIS Indicative Range description:        Between MSL and HAT (upper intertidal)

### 3.2.5 Indicator Types: Salt marsh (various subtypes)

Salt marshes have been used extensively in Holocene sea level research, as their elevation and position is directly controlled by the tidal elevation and therefore can be directly related to a reference water level (Engelhart and Horton, 2012; Shennan et al., 2018; Barlow et al., 2013).  The indicator type differ from 'basal peat' and 'estuarine tidal flat' types, because study of

microfossils allowed to identify it as a coastal salt marsh specifically. The identification of palaeo salt marsh is usually through the identification of salt marsh specific taxa such as the pollen of *Plantago maritima* and *Triglochin* (Gehrels, 1994);





the presence of salt marsh foraminifera such as *Jadammina macrescens, Miliammina fusca* and *Trochammina inflata* (Gehrels, 2000; Edwards and Horton, 2000); and brackish water diatoms and ostracods (Penney, 1987; Barlow et al., 2013; Zong and Horton, 1998). Such microfossils cover a limited elevation range from HAT to MSL and hence make the salt

marsh provide an excellent sea-level index point. Where sedimentation could keep up with the rate of RSL change, salt marsh may be preserved at the transgressive or regressive boundaries often between freshwater peats and estuarine silts and clays. Microfossil sampling resolution is often coarse in LIG estuarine sediments (>5-10 cm intervals) and coastal salt marsh deposits may be missed between samples. Therefore, there are only two explicit occurrences in the LIG salt marsh SLIPs within the NW European database. A number of sites classified as 'basal peats' may well be overlain by fine grained

marine sediments that could be salt marsh. Also several 'estuarine terrace tidal flats' sites are reported to have been overlain by salt marsh muds and peats (e.g. Balgerhoeke, Belgium, IDs 339 and 340; Land Hadeln, Germany. IDs 882 and 883). Revisiting sites of these type may present opportunities to narrow down the indicative range of some SLIPs.

In formula:

WALIS Relative Water Level description:   (HAT to MSL)/2

WALIS Indicative Range description:        Highest astronomical tide - mean tide (or sea) level

### 3.2.6   Indicator Type: Shallow or intertidal marine fauna

This generic SLIP indicator type entry in WALIS considers palaeobiological identified marine fauna that can be associated with very shallow water and/or intertidal environments, especially where fossilized 'in viva'. It was used when sedimentary, morphological identifiers are not available, and where sites did not convincingly fall in one of the categories listed above. In

the North Sea, macroscopically, usually these are the shells of intertidal mollusca. In Danish contexts, also foraminifera assemblages have been used as water depth and current regime biotic indicators in Eemian shallow marine beds (e.g. Konradi 1976, Konradi et al., 2005), as are diatom assemblages (e.g. Van Leeuwen et al., 2000; Beets et al., 2008). Most commonly, though, it is the macroscopically spotted shells of molluscan fauna that are used. *Cerastoderma edule* (also known as *Cardium*) and *Scrobicularia plana* are very common intertidal species in the North Sea, in Holocene and LIG

deposits alike. *Macoma Baltica* and *Spisula truncata* are also frequently encountered (and used for AAR characterisation; Miller and Mangerud 1986, Meijer and Cleveringa, 2009, Demarchi et al., 2011). In deeper waters of the offshore North Sea, Skagerrak and SW Baltic, the common species are *Arctica islandica* (also known as *Cyprina*) and *Turritella communis*. In the English Channel, foraminifera *Elphidium* sp. and *Ammonia* sp. are common intertidal indicators (e.g., Bates, 2010). Some of these species ('Lusitanian components') in the LIG extended their common presence into North Sea and SW Baltic

too (e.g. Madsen et al., 1908; Miller and Mangerud, 1986; Funder et al., 2002; Meng et al., 2021), whereas in the Holocene they did not (or to a far more limited extent). Examples are *Venerupus senescens* (*Tapes aurea (var. eemensis), Paphia aurea, Amygdala*), *Bittium sp., Cardium exiguum*. Nolf (1973), Miller and Mangerud (1986: their part II), Meijer and Preece (1995), Wesselingh et al. (2010), Meijer et al. (2021) are illustrative biota-oriented papers on this. This means that literature-reported molluscan faunas hold both 'vertical' indicative meaning, as well as 'age' chronostratigraphic meaning.





In formula:

WALIS Relative Water Level description:  Based on the upper and lower limits of living modern analogue faunas

WALIS Indicative Range description:      Based on the upper and lower limits of living modern analogue faunas

**3.3      Chronostratigraphical entries**

The Dutch, Belgian, German and Danish sites are primarily studied through pollen-zone biostratigraphy, which is also used
as relative dating information (e.g. Lambeck et al., 2006; Kopp et al., 2009: "relative ages are known with more precision
than absolute ones"). Hence, in the WALIS database, we defined 'chronostratigraphic method' standardized entries (Table 4)
to encode this form of dating control, and used these for the Netherlands', Belgian and German Bight sites. The standard
entries refer to the six main pollen zones of the Zagwijn (1996) floating scheme, and the descriptions include the correlated
schemes of Zagwijn (1961, 1996) for the Netherlands; Behre (1962), Selle (1962), Müller (1974) and Menke and Tynne
(1984) for NW Germany; Jessen and Millers (1928) in Denmark and NW Germany; and further discussions in Benda and
Schneekloth (1965), Grüger (1989),  Zagwijn (1969), and Funder et al. (2002). Abundant literature and ongoing debate exist
regarding absolute timing of onset of the LIG and degree of diachronicity within and between the regional schemes, their
correlation with Southern Europe counterparts, and (non-)analogy with the Lateglacial-Holocene (e.g., Van Leeuwen et al.,
2000; Cleveringa et al., 2000; Turner, 2002; Kukla et al., 2002; Beets and Beets, 2003; Beets et al., 2006; Lambeck et al.,
2006; Sier et al., 2015; Long et al., 2015).

To make database usage not too dependent on shifts and positions in debates on timing and duration of the NW European
Eemian (covered in Section 6.1), the design of the 'Dated By: Chronostratigraphy' entries (Rovere et al., 2020) requires
filling four duration and timing database fields.  Two of these deal with the palynological correlation and varve-counting
locally established durations and its uncertainty (Müller, 1974; Zagwijn, 1996 – see Table 4 footer). Two further fields store
a lower and upper limiting numeric age. The first one bounds the onset of the interval (oldest moment the zone may have
begun), the second one the ending (latest moment represented in the zone). For a fictive zone with a duration of 4000 years
that could be perfectly pinpointed in time, lower and upper limit would be 4 ka apart. For a floating chronology such as the
set of zones used in the North Sea region, the limits are some 10 ka apart (e.g. the PAZ E5 and E6 rows in Table 4), the
difference being the 6 kyr difference entertained for the onset of the Eemian in the literature of the last 20 years (see Long et
al., 2015 for summary). Entries were made for 23 LIG chronostratigraphic divisions (Table 4). 18 of them dealing with
terrestrial palynological subdivisions (see Table 4 footer); 5 dealing with SW Baltic and Kattegat-Skagerrak marine
environmental phases (Cyprina clay phases of Kristensen et al., 2000; Funder et al., 2002; Kattegat stadial of Seidenkranz,
1993), that in turn were correlated and connected to the terrestrial schemes. A meta-field storing parent-child relationships
for chronostratigraphic division was also maintained. Comprehensive explanatory text, referencing positions taken in
literature, fills the note field for the parent entries and provide a  rationale for the lower and upper numeric ages provided.





**Table 4:** 'Dated by: Chronostratigraphy' entries for North Sea, Skagerrak-Kattegat and SW Baltic Sea

| Literature derived age constraining information, encoded in WALIS' chronostratigraphic unit duration and age fields | | | | | Scenario-based age attributions, fitting WALIS' Lower and Upper constraints | | |
|---|---|---|---|---|---|---|---|
| Chronostratigraphic divisions tied to North Sea LIG sea-level data points | Duration constraint (yrs)** | Duration uncertainty (yrs)*** | Lower numeric age (ka) | Upper numeric age (ka) | Suggested oldest possible *onset* (ka) | Middle of the road (ka) | Suggested latest possible *ending* (ka) |
| *Zagwijn 1961, 1983, 1996 \** | | | | | | | |
| *EW-I* | *10,000* | 800 | 116 | 102 | 116 to 105 | 113-103 | 110 to 102 |
| *E6/EW break* | *( 600 )* | 200 | 116 | 110 | ~116 | ~113 | ~110 |
| E6 | 4000 | 250 | 120 | 110 | 120 to 116 | 117 to 113 | 114 to 110 |
| (E6a, E6b) | (2000, 2000) | (180, 180) | | | | | |
| *E5/6 break* | *( 600 )* | 200 | 120 | 114 | ~120 | ~117 | ~114 |
| E5 | 4000 | 250 | 124 | 115 | 124 to 120 | 121.5 to 117.5 | 119 to 115 |
| E4 | 1800 | 110 | 125 | 119 | 125 to 123 | 123 to 121 | 121 to 119 |
| (E4a, E4b) | (700, 1100) | (50, 100) | | | | | |
| E3 | 675 | 45 | 126 | 121 | 126 to 125.3 | 123.7 to 123.2 | 121.5 to 120.8 |
| (E3a, E3b) | (225, 450) | (25, 35) | | | | | |
| E2 | 425 | 30 | 126 | 122 | 126.5 to 126.0 | 124.7 to 124.2 | 122 to 121.5 |
| (E2a, E2b) | (200, 225) | (20, 25) | | | | | |
| E1 | 100 | 10 | 127 | 122 | ~126.5 | ~124.5 | ~122 |
| Entire Eemian (E1-E6) | 11000 | 500 | 127 | 110 | 127 to 116 | 124 to 113 | 121 to 110 |
| *Seidenkrantz 1993* | | | | | | | |
| Kattegat Stadial | *1200* | 200 | 128 | 122 | 128 to 126.5 | 125.6 to 124.4 | 123.5 to 122.0 |
| *Kristensen et al., 2000* | | | | | | | |
| Cyprina clay (undivided) | 6750 | 500 | 126 | 114 | 126 to 119 | 123.5 to 116.5 | 121 to 114 |
| Cyprina clay, upper saline (E4b, E5, E6a)* | 4700 | 400 | 124 | 114 | 124.0 to 119.3 | 123.2 to 117.5 | 119 to 115 |
| Cyprina clay, lower saline (E3b, E4a, E4b)* | 1950 | 200 | 126 | 119 | 126 to 124 | 123.5 to 121.5 | 121 to 119 |
| Cyprina clay, lowest brackish (E2b, E3a)* | 350 | 100 | 126 | 121 | 126.2 to 125.8 | 123.7 to 123.6 | 121.4 to 121.0 |

\* WALIS entries use Zagwijn (1961, 1983, 1996) scheme. Descriptions for entries include correlations to schemes of: Jessen and Milthers (1928, op. cit. Funder et al., 2002:Table 2); Selle (1962 op. cit. Konradi et al., 2005:section 8); Behre (1962); Müller (1974) and Menke and Tynni (1984). Correlation for Cyprina Clay complies to that in Kristensen et al. (2002) and Meng et al. (2021).

\*\* In normal font, for E1 to E6: based on Müller (1974), Zagwijn (1996), NW-German low land varved lake site 'Bispingen'. In *italics*: For E5/6 break and E6/EW break: arbitrarily defined to 300 years on either side of break. For EW-I: compliant varve counts in Eifel mar lakes (Sirrocko et al., 2005) and their dust-flux correlation with Greenland ice-cores NGRIP and NEEM (ibid., NEEM community members, 2013; Sier et al., 2015). For Kattegat Stadial: assessment of Seidenkrantz (1993).

\*\*\* Author expert judgement: acknowledges varve miscount possibilities as well as PAZ correlation diachronicity issues over ca. 400 km distance between 'coastal Netherlands' (Zagwijn, 1961; Amersfoort) and 'NW German terrestrial' sites (Müller, 1974; Bispingen) and/or Eifel mar lakes, and/or NE Denmark and/or the SW Baltic.



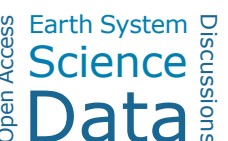

Section 6.1 includes reasoning and reference on why to enter 127 kyr (a few 1000 years into MIS5e) as the oldest advised onset age for NW European Eemian pollen zones, and not 130 kyr (midpoint Termination II in marine oxygen isotope stacks). The geochronological user of the database may want to extract PAZ durations and suggested age-midpoints from WALIS (its equivalent of Table 4), formulate some rules that make sure that PAZ order and cumulative duration remain honoured while shifting the PAZ-series to earlier or later moments in time (e.g. Bronk-Ramsey, 2008), and then invoke

Bayesian age-modelling approaches to explore what amount and direction of shift best to minimizes differences with sites with numerical age control. This is a challenge, however, because the absolute dating techniques may themselves be subtly biased, to older or younger side. Ideally, a regional Bayesian calibration approach combines the optimizations for in the vertical (RSL) and temporal (age) directions (e.g. Hijma and Cohen, 2019). The NW Europe database and that of WALIS more broadly may inspire such work for the LIG (Düsterhus et al., 2016).


We stress that the numeric ages and durations advised are for the North Sea region mainly. Site Bispingen where the Müller-1974 varve-counts come from (Fig. 1), lies within 100 km of sites in the German Bight and within 500 km of the Belgian sites where in our dataset the scheme was applied furthest away. If WALIS entries for RSL indicators with palynology-derived chronostratigraphic age control are needed for sites further east (e.g. where the Russian 'Mikulinian' zonation

schemes are used) or to the southwest in the English Channel (see next section), we advise to make new separate entries in WALIS, allowing to adapt advised oldest and youngest age bounds, and stimulating users to explore possible W-E diachronicities.

## 4    Elevation measurements and corrections

### 4.1 Present elevation, national datums

The measurement elevations for the studied sites been reported in various forms (Table 5). Our approach has been to express data points as much as possible to Datums relating to 19/20[th] century reference mean sea level (Table 6), applying conversions when needed. Hereto, we checked and registered against which datum the elevations were expressed in original studies (for further dicussion see Woodroffe and Barlow, 2015). Some countries in the study area use a national datum that approximates modern mean sea level elevation (e.g. Netherlands: NAP, German: NHN and NN, Danish: DNN and DVR90;

e.g. Wiśniewski et al., 2014). In these cases we express the elevations against national datum (typical entries then reproduce originally reported values). Other countries use a national datum that approximates low mean springtide / lowest astronomical tide (e.g. Belgium: TAW and UK: regional Chart Datums). Being larger countries, the relationship between the national datums in the British Isles and France (e.g., the UK's Ordnance Datum, OD) and low tide and mean sea-level is geographically more variable (Bradshaw et al., 2016; Bradley et al., 2011). For Belgium, the approach has been to subtract

2.33 m from reported elevations expressed to TAW datum (Table 6) to express elevations to a MSL based datum, as in surrounding countries.





**Table 5: Measurement techniques used to establish the present elevation of Last Interglacial Sea-level data points**

| Measurement technique | Description | Typical accuracy |
|---|---|---|
| Cross-section from publication | The elevation was extracted from a published sketch/topographic section. | Variable, depending on the scale of the sketch or topographic section. |
| Differential GPS (DGPS) | Positions acquired in the field by 'rover' GPS stations, corrected either in real time or during post-processing with respect to the known position of a 'base' GPS station (or a geostationary satellite system). DGPS accuracy depends on distance from base station, and number of static positions acquired per location. | ±0.02/±0.08 m, depending on survey conditions (e.g. satellite reception) and instruments used (e.g., single-band vs dual-band receivers) |
| Metered tape or rod | The end of a tape or rod is placed at a known elevation point, and the elevation of the unknown point is calculated using the metered scale and, if necessary, clinometers to calculate angles. | Up to ±10% of elevation measurement |
| Total station or Auto/hand level | The accuracy of the elevation measurement is also inversely proportional to the distance between the instrument and the point being measured. Furthermore, it takes over the accuracy of the benchmark used when setting up the Total Station or Hand levelling survey. | ±0.1/±0.2m for total station<br><br>±0.2/±0.4 m for hand level |
| Multibeam bathymetry data + core depth | Bathymetry derived from multibeam surveys in offshore areas, below which the depth along cores is expressed. Errors differ with coring system (gravity, vibrocoring, rotary drilling...) and should be assessed case by case. | The accuracy of modern MBES systems is considered of cm-scale (<0.1 m), with main errors derived from tidal correction during the survey day. Core depth error sources are: core shortening, non-vertical drilling, compaction. |
| Not reported | The elevation measurement technique was not reported, most probably hand level or metered tape. | 20% of the original elevation reported added in root mean square to the sea level datum error |

For the British Isles, local reference water levels are calculated at 44 tide gauge locations based upon multiple decades of direct measurement. This establishes the regional Chart Datums, that are used to assess the indicative range of a local modern analogues (e.g. beach crest height). Elevations reported to regional UK datums were converted to the MSL-based national OD at Newlyn (using values given by the National Oceanography Centre; ntslf.org/tides/datum). Sea level at Newlyn has risen ~0.22 m above ODN since first established between 1915-1921. Modern MSL at Portsmouth (southern England) and Sheerness (southeastern England) is ~ +0.14 m ODN and at Immingham (eastern England) ~-0.23 m ODN (Holgate et al., 2013; PSMSL, 2021).





For France, all elevations are given relative to NGF-IGN69, with the 0 level mean sea level in Marseille in the Mediterranean (Table 6). MSL along the French side of the Channel (SHOM, 2020) is at +0.5 m IGN69 for Brittany and approaches +0.6 m in the Strait of Dover (at present, 0.10-0.15 lower in early 20[th] cy based on tide gauge data; Wöppelmann et al., 2008). The user of WALIS data may thus opt to lower French-datum related elevations by c. 0.4-0.5 m prior to entering analysis. The above geographically generalized datum-related conversions are pragmatic ones. We estimate the systematic vertical error associated to be 0.05-0.20 m as other sources of systematic and stochastic error in the calculation of SLIPs from the LIG are regarded greater, this approach is warranted.

**Table 6: Sea level datums made use of in this study.**

| Datum name | Datum description |
|---|---|
| Mean Sea Level / General definition | General definition of MSL, with no indications to which datum the measurement referred to. A datum uncertainty can be established on a case-by-case basis. |
| DNN (Danish ordnance datum, prior to DVR90) | DNN (Dansk Normaal Nul) is the O.D. in Denmark used during the 20[th] cy. It is about equal to second-half 20[th]-cy MSL (mean half-tide) as observed in 10 tide gauges along the Danish coast. DVR90, which ties to NAP, replaced DNN in the early 2000s. 0.02 m DNN is 0 m DVR90 in N Denmark; -0.14 m DNN is 0 m DVR90 in SW Denmark. |
| NAP, NHN, NN (Netherlands ordnance datum, Amsterdam, also 0 level for the German ordnance datum) | NAP (Normaal Amsterdams Peil) is the O.D. in the Netherlands, and NHN (Normalhöhennull) the German O.D. that shares the datum. The zero level is about equal to second-half 20th cy MSL (mean half-tide) as observed in tide gauges along the Dutch coast. NHN replaced precursor NN in the early 2000s, that also tied to NAP. |
| TAW+2.33 (Belgian ordnance datum, Oostende, offset to get from LAT to MSL) | TAW is a datum based on Lowest Astronomical Tide (LAT) at Oostende (Belgium). 0 m TAW (Belgium) is -2.33 m NAP/NHN (Netherlands, Germany), and hence +2.33 m is the datum offset. For reference: 0 m TAW (Belgium) is -1.83 m NGF (France). |
| OD (British Ordnance Datum Newlyn) | In the U.K., OD is defined as the mean sea level at Newlyn (Cornwall, UK) between 1915 and 1921. For reference, modern MSL is 0.14 m above ODN at Portsmouth and Sheerness, and 0.23 m below ODN at Immingham. |
| NGF-IGN69 (French Ordnance Datum, Marseille) | NGF (Nivellement General de la France) is the ordnance datum for continental France. The zero level equates to Mediterranean mean sea level (mean half-tide) in Marseille as gauged between 1885 and 1897. For reference, 20/21[th]-cy MSL is 0.505 m above NGF at Brest, 0.585 at Cherbourg, 0.491 at St Malo, 0.585 at Le Havre and 0.571 at Calais, and 0.50 m NGF at Oostende (Belgium, see TAW). |



## 4.2 Compaction corrections

Most SLIPs and limiting data points are from sites and indicator types that by their nature do *not* require particular
compaction correction e.g., marine terraces, beach deposits, drowned valley floor and isolation basin sills (Table 3). Data
points where compaction corrections have been applied are mainly to those collected from boreholes in the Netherlands, NW
German and SW Danish settings that sampled basal peat, lagoonal and estuarine tidal flat type indicators. In those cases,
compaction was assessed for organic and muddy beds.

For basal peats we assume full analogy with Holocene deposits. Where Holocene peats are overlain by 10-15 m of coastal
overburden, transgressed peats are compressed to 50-33% of their original thickness, with the greatest majority of this
compression happening in the first millennia after burial (Hijma et al., 2009; Hijma and Cohen, 2011). Decompaction of
decimetre-thick basal peat beds is achieved by multiplying thickness by a factor 2 to 3 (Berendsen et al., 2007; Hijma and
Cohen 2019). For Eemian basal peats, the overburden is similarly thick, but in-situ for longer time and relatively sandy. We
hence considered a decompaction factor of 2.5 to 3.5 for dominantly organic beds, and 2 to 3 for clay-peat alternating
intervals. As basal peat SLIPs and associated marine limiting indicators lie at the top of a basal peat bed (§3.2.6) this affects
the elevations of data points of these types.  For lagoonal and estuarine tidal flat tops, decompaction of clayey tidal deposits
immediately underlying these levels is considered. It is only 'post-depositional' compaction after the tidal flat ceased to
function that is to be assessed in that case; not syn-depositional 'auto-compaction' or the loading and lowering of underlying
basal peaty strata as that was accommodated while the tidal deposits accumulated (Brain, 2015). Subtidal clayey facies is
more prone to post-depositional compaction (decompaction factor 1.5 to 2.5) than intertidal and supratidal facies
(decompaction factor 1 to 1.5) because the latter had been subjected to wet-dry cycles at time of deposition (Paul et al.,
2005). The amount and composition of overburden affects opting for the lower or higher side of decompaction.

The above types of compaction corrections are documented in the 'Notes on elevation and indicative range' field in WALIS,
for each entry where it was applied. The decompaction approach was a pragmatic one. Geomechanic empirical-calibrated
modelling suggests to similar decompaction factors for basal organic beds and clayey tidal lagoonal sequences (e.g.
Greenfield and Tucker, 1986; Brain, 2015), as long as the post-depositional compaction component is isolated from the syn-
depositional one.


A few sites in the database are from particularly thick organo-clastic Eemian sequences, from deglaciation-inherited deep
channels onshore in the German Bight (e.g. Dagebuell, Schnittlohe) and from glacial-tongue basins in the Netherlands (e.g.
Amsterdam, Amersfoort; Fig. 1). We avoided defining heavily-compaction influenced sea-level markers and regression
terrestrial limiting points from such localities. From the deep and thick German Bight sequences, SLIPs and limiting points
are only defined based upon deepest levels in the sequences, for which compaction corrections are relatively minor. The

environments became subaqueous following the transgression and it is the proxy water depth inaccuracy, rather than decompaction, that affect indicator elevation position and associated uncertainty. For the Amersfoort tongue basin, the database provides additional locations close to the rims of the basins as more accurate highstand and regression indicators. For the Amsterdam tongue basin transgressive information, we define the locality as an insolation basin, as the elevation
information comes from a sill in glaciogenic substrate where compaction can be ignored.

### 4.3  Vertical Land Motion (VLM) correction

VLM correction values are supplied for the North Sea Basin depocenter (Fig. 1), but not for the other regions covered in this paper. We focus on the North Sea Basin due to both its significance and precedent. Firstly, rates of VLM in the North Sea Basin are in the order of 0.1-0.2 m/kyr, meaning up to 10 meter of correction in the regions of the LIG highstand shoreline,
and up to 20 meters further offshore at locations closer to depocentres. As noted in section 2.1, that spatial variation in tectonic subsidence affects how transgressive, stillstand and regressive RSLs are to be compared.  Key literature compiling and using LIG sea-level datasets for Europe and globally hence have earlier applied basin-subsidence VLM values to the North Sea Basin sites (Lambeck et al., 2006; Kopp et al., 2009). Including assessed VLM values in WALIS entries allows application of such corrections to dataset to be consistent, spatially across the covered region as well as with the previous
studies (for data points then included).

Areas around the margin of the North Sea Basin are thought to experience relatively minor non-GIA VLM. Very modest long-term subsidence rates may apply to NE Belgium, Denmark and N Germany during the Pleistocene, or alternatively they can be viewed as fairly stable (Kiden et al., 2002). Whether the Dover-Calais area is neotectonically active, and owing to what cause, is debated (e.g. Van Vliet-Lanöe et al., 2000; Westaway et al., 2002; Garcia-Moreno et al., 2015). The region is
also known to have lost considerable volume of bedrock in Middle Pleistocene times (proglacial erosion of Dover Strait; Gibbard, 1995; Gupta et al., 2007, 2017) with some isostatic uplift in response. Regardless, the apparent uplift due to both GIA and non-GIA VLM based upon the flight of raised beaches is modest (+0.02 to +0.04 m/kyr; Pedoja et al., 2018). Similar applies to sites along the English and French sides of the English Channel. The flight of raised beaches in West Sussex (Bates et al., 2010; Briant et al., 2019) occurs in an area where net uplift is thought to be significant, with apparent
rates between 0.06 and 0.12 m/kyr, explaining the mean vertical separation of highstand beaches of the last few cycles (e.g. Westaway et al., 2006). However, this rate calculation is sensitive to age-attribution (sections 2.5-2.7) and assumptions on global ice sheet history and spatio-temporal variations in the solid Earth, and again a breaking down the relative contribution of different process of uplift is complex. As a result, we do not provide a VLM value for these regions. Independent constraints on long-term VLM, separate from GIA, will be an ongoing challenge in this area.


Background VLM rates characterizing non-GIA subsidence in the southern North Sea are obtained from Kooi et al. (1998: their Figure 4, all three components totaled). They result from a tectono-sedimentary back-stripping analysis on thickness of





Quaternary and Neogene sequence of the basin and presented mean rates estimated over the last 2.6 Ma. The starting point for that subsidence analysis is the mapping of Quaternary thickness in the North Sea Basin, based on offshore seismo-stratigraphic and onshore lithostratigraphic mappings, collated. For reanalysis and assessment of the 1998 outcomes, we generated such source materials for The Netherlands and direct vicinity with current onshore-offshore geological survey digital mapping resources, and are able to broadly reproduce the accommodation patterns and subsidence rates Kooi et al. (1998). Further offshore, mapping of the base Quaternary has been revised (e.g. Lamb et al., 2018). This shifted depocenter contours, which locally nudged VLM values with -0.03 to +0.03 m/kyr relative to the 1998 results. The contour lines in Fig. 4 served as VLM-subsidence isolines where values for newer data points had to be assessed. VLM used in WALIS for this region ranges from 0.02 m/kyr at marginal locations, to 0.24 m/kyr at sites over Quaternary depocentres. VLM rate uncertainty is ±0.01 for sites along the basin margin (subsidence of -0.02 to -0.03 m/kyr) and increases to ±0.04 in far offshore depocentres.

There is an indication (analysis in Barthes et al., 1999; Kuhlman et al., 2006ab and Arfai et al., 2018) that the 2.6-0 Myr averaged subsidence rate, is controlled by very strong sediment-loading subsidence induced between 2.6 and 1.8 Ma (rates higher than average), followed by much reduced subsidence after 1.8 Ma (rates below average, estimated at 80%). If so, the VLM value we use in WALIS should be regarded as maximum background subsidence rate. Breaks in basin subsidence trends before and after 1 Ma may be suspected for the southern North Sea, but have not been spatially quantified. Caveats of glacial-interglacial variations in the height of regional (what mean value to use, given GIA overprinting) make estimating accommodation-based long-term subsidence rates difficult for this youngest period. For this reason, the WALIS VLM-rates for the North Sea Basin relies of identification of surfaces from Early Pleistocene marine basin, below the glaciogenic 'upper regional unconformity' (cf. Ottensen et al., 2014).

Where we report VLM-corrected elevations for North Sea SLIPs and limiting data points in the next section, we calculated these with the midpoint numeric ages as listed in Table 4. Such projected elevations rise if points are shifted to the older side, and drop if they are shifted to younger side of the considered range. In the transect plot of Fig. 5, such will affect transgression and regression points more than highstand points (see above). For age-depth plots effects will be considerable. The uncertainties we specified for the VLM rates, propagate to vertical uncertainties when applying the VLM correction. Doing so for the Zagwijn-1983 subset, reproduces the magnitudes of uncertainty in tables and plots in Lambeck et al. (2006). The rate uncertainty specified for North Sea offshore site '#2' is similar to that used in Kopp et al. (2009): -0.03 vs -0.0255 m/kyr. For the inshore sites, the rate uncertainty specified in WALIS, is roughly twice the value in Kopp et al. (2009).

![North Sea Basin background subsidence (VLM) map showing Pleistocene depocentre with provisional VLM isolines and values registered for data points]

**Figure 4: VLM values (m/kyr) for North Sea Basin LIG datapoints.** Area of subsidence and VLM contours, based on depth base-Pleistocene in marine formations (onshore and offshore data NL, GER and DK geological surveys), negotiated for water depth reduction during the Early Pleistocene (isoline increment: 0.04 m/kyr, i.e. equivalent to 5 m RSL correction accumulated over 120 kyr). Values in regular font: WALIS VLM entries for LIG data points, based on Kooi et al. (1998). Values in italics: extrapolations thereof (this paper). Offshore, the subsidence patterns shown deviate subtly from those in Kooi et al. (1998), because of improved mapping of base Pleistocene. Backdrop as in Figure 1, contains land data © OpenStreetMap contributors 2020, distributed under the Open Data Commons Open Database License (ODbL) v1.0.



## 5 Overview of data points

In this section we present an overview of data points, that for graphic representation and textual attention are grouped in:

- a main cluster coming from the 'highstand shorelines', the term used here in sequence stratigraphic rather than RSL sense, defining a general stillstand reached within the interglacial.
- a 'transgression' cluster with relatively older and relatively deeper positioned RSL data points, from the rising limb towards the highstand / stillstand;
- a 'regression' cluster, with relatively younger RSL data points, that come from positions seawards of the highstand cluster and from relatively lower elevations.

From each of these groups, SLIPs as well as marine limiting and terrestrial limiting points are present in the database, as Figures 5 and 6 show for the North Sea and Channel regions. These figures are transect plots with locations projected to a central axis. Datapoints affected by North Sea basin subsidence are plotted twice: VLM-corrected in colour, raw in gray (numeric age used to calculate correction: Table 4, middle option; subsidence rates in Fig. 4).

Lithostratigraphic (architectural, cross-sectional relationships) and biostratigraphic (regionally reproducing biogeographical successions) relative dating evidence is used to distinguish between the groups. In areas with sites where lithostratigraphic and biostratigraphic evidence is ambiguous or of coarse resolution, we place the data point in to the 'stillstand' group, to when it is not known if they are from a regressive or transgressive stage. This mainly applies to stretches of French and English coast. Sites along the Belgian, Netherlands', German and the SW Danish coast and offshore, are generally resolvable into transgression, highstand and regression phases. From Belgium to the German Bight, a stillstand is perceived during PAZ E5 (~4000 yrs long, Table 3), and regression commences in PAZ E6. Therefore, the chronostratigraphic coding determined our grouping in this region, with points labelled as 'PAZ E5/6 break' the youngest ones in the stillstand group. In N Denmark and the SW Baltic the stillstand also straddles PAZ E5, but is less precisely constrained (Cyprina Clay upper saline phase; E4b-E6a) and hence may have lasted a little longer (~4700 yrs; Table 3). Dividing the transgression and stillstand (highstand), from the perspective of continuous SLR slowing down or a difference in elevation was done in an arbitrary matter: all data points of Late Saalian to PAZ E4b chronostratigraphic age were put in the 'transgression' group (Table 1).

Figure 5 shows even numbers of 'highstand' and 'transgression' data points for the North Sea region. The English Channel coastal data in Figure 6 predominantly contains stillstand datapoints. For reference: the ~4000 years of PAZ E5-equated stillstand time (North Sea) will fall within the middle and/or the second half of MIS 5e, whereas the regression onset (equated to beginning of PAZ E6) may be toward the end of MIS 5e or into MIS 5d (options in Table 4). Across the English Channel region, the regional duration of the more broadly defined stillstand is not particularly resolved, but a stillstand in the second half of MIS 5e may be expected. The further overview is organised by region.







**Figure 5: SW-NE organised plot of Last Interglacial WALIS data entries for the North Sea, split to type and relative position.**
**Transect location in Fig. 1. Bottom panels shows sites and age-depth plot as a screenshot of WALIS' interactive viewer (Garzon**
**and Rovere, 2021): https://warmcoasts.eu/world-atlas.html, which includes © OpenStreetMap contributors 2020 data as basemap**
**(distributed under ODbL 1.0). Only LIG data points plotted.**





## 5.1 North Sea: Netherlands and Belgium

For the Netherlands and Belgium, availability of national online geological datasets (e.g. Van der Meulen et al., 2013), means that all sites and boreholes in this region were looked up in web-portals to verify coordinates, surface elevations, layer depths etc. (Fig. 2b). For The Netherlands' we used www.dinoloket.nl/en/subsurface-data, for Belgium (Flanders)
www.dov.vlaanderen.be.  The WALIS entries include borehole IDs from these portals. Coordinates of legacy sites based on outcrops were also checked using digital topographic and aerial photography resources.

Figure 1 in the North Sea depicts the Eemian highstand coastal situation where the inner rim of the coastal plain featured deep incursions where river valleys and glaciogenic basins and outwash channels were marine inundated, dented with
morainic headlands. North Sea highstand wave action during the Eemian would reshape these headlands, while long-term basin subsidence would lower their absolute elevation. Falling stage and lowstand terrestrial sedimentary process (river valley developments in the basin; Busschers et al., 2007; Peeters et al., 2016) eroded most Eemian coastal plain features (Peeters et al., 2019) .  Locally terrestrial peats bearing late interglacial and early glacial palynology (PAZs E6 and EW in Table 1) formed over highstand-originating surfaces that preserved along the rims of lagoons and estuaries preserving
regressive stage RSL data points.  Local preservation tends to be explained by burial under periglacial colluvial-alluvial deposits from the onset of the Last Glacial (Early Weichselian, Table 1), besides the long-term subsidence.

Locally, deposits along the rim of the Eemian highstand coastal plain have preserved as lagoon- and estuary-fringe sites in the central Netherlands, most famously Amersfoort (Zagwijn, 1983; Cleveringa et al., 2000; WALIS IDs 133, 137, 138), but
also at Oosterwolde (Peeters et al., 2016; IDs 134-136), Rutten (Sier et al., 2015, Peeters et al., 2016; IDs 109, 111, 114, 115), Scharnegoutum (Zagwijn, 1983; IDs 124-127) and Annen (Bosch, 1990, IDs 4260, 4261). These sites each provided multiple SLIPS and limiting points, including the final transgressive (all PAZ E4) and early regressive stages (all PAZ E6; site Oosterwolde OSL-dated). The inland-most coastal beds from the highstand ('stillstand') stage are nowadays encountered 8 m below MSL.  With VLM-corrections, 'highstand' SLIPs (from PAZ E5) from Amersfoort plot at +3.5 m MSL, and the
full cloud of highstand SLIPs projects between +1.2 m (Annen) and +6.2 m MSL (Scharnegoutum). Four transgression phase SLIPs coming from these sites have paleoRSL depths between -21 and -13 m MSL. With VLM-correction their midpoints plot between -8.3 and -1.4 m (PAZ E4a) and between -7.1 and +1.2 m (PAZ E4b). SLIP uncertainties range ±0.35-1.3 m, with decompaction uncertainties included. Propagating the VLM-correction uncertainties (±0.030 m/kyr) enlarges the error to ±3.6-4.5 m (Figure 5).

Seaward from the Eemian coastal plain inner rim, further marine deposits are encountered but at greater depths down to $^-$40 m below the coastal plain, and at -35 to -65 m MSL offshore. This area provides SLIPs from during (initial) transgressive stages (coincident with PAZ E3) such as near Amsterdam (Zagwijn, 1983; Cleveringa et al., 2000; De Gans et al., 2000; Van





Leeuwen et al., 2000; Beets and Beets, 2003; IDs 118-120) and offshore boreholes 'BH89/2' (Sha et al,, 1991; Beets et al., 2005), as well as during (developed) regressive stages (coincident with PAZs E6 and EW), such as at Petten (Zagwijn, 1983: IDs 146-148). The transgressive sites also provide critical terrestrial limiting datapoints for the transgression (PAZ E1-E2), and less critical marine limiting points from during and after transgression (PAZ E4, E5) because these covered more accurately by the inland sites described above. The transgressive SLIP from Amsterdam (ID 118) is an 'isolation basin (marine connection)'. The sill-height based SLIP (PAZ E3a) occurs at -39.0 ± 2.0 m MSL (tidal amplitude and sill elevation uncertainties considered) and increases to -19.4 ± 4.8 m MSL with VLM-correction. Offshore site BH89/2 is also a glacial basin (Beets et al., 2005), and an isolation basin-type transgression-stage SLIP has been registered (ID 122; PAZ E3a) at -53.8 ± 9.0 m MSL, where the large uncertainty comes from poorly defined sill location. This is the site with greatest estimated basin subsidence (Fig. 4). Applying VLM of -0.24 ± 0.05 m plots the point at -24.1 ± 10.9 m MSL (Fig. 5). The regressive SLIP for Petten (ID 148; PAZ E6b) is constructed by combining coeval terrestrial and marine limiting evidence in a transect of cores (Zagwijn, 1983), and plots at +0.9 ± 4.7 m (VLM corrected; Fig. 5). North Sea boreholes '#2', '#3', '#5' and '#9' (Zagwijn, 1983) provide terrestrial and marine limiting datapoints only (IDs 309, 311, 317, 319  320, 380), as do onshore boreholes such as Noorderhoeve (ID 898; Meijer, 2002; Peeters et al., 2016).

The majority of Belgian RSL data points are estuarine tidal deposits preserved as a W-E series along the rim of the LIG Scheldt estuary (Fig. 1 and Fig 2d). They occur at relative shallow subsurface positions than in the Netherlands, with three inland SLIPs coming from pit exposures: Meetkerke (De Moor and De Breuck, 1973; Nolf, 1973; ID 3259 – see Fig 2e), Pit Dhondt and Pit Coppens (Heyse 1979; IDs 341, 345), and three more from boreholes at Waterpolder and Balgerhoeke (Heyse, 1979; IDs 338-340). The latter is the most eastward and shallowest site, preserving highstand supratidal mud (ID 339, PAZ E5) and salt-marsh peat (ID 340, PAZ E5). The other SLIPs are from intertidal levels. For PAZ E5, the NW Belgian RSL positions top between -3.1 and -2.9 m MSL (Pit Coppens, Waterpolder, Balgerhoeke). A value of -5.4 m MSL (Pit Dhondt) also associated to PAZ E5 may reflect late highstand/start of the regression. These values incorporated large modern-day outer estuary tidal amplitudes for the Scheldt, and applying no VLM. The Belgian mesotidal SLIPs have paleoRSL uncertainties ranging ±0.8-1.6 m. Reducing tidal range and considering modest subsidence (e.g. 0.02 ± 0.02 m/kyr) will raise the elevation of the Belgian points and warrants further investigation.

Westwards, the sites reveal truncated sequences lacking supratidal contacts, generating transgressive SLIPs from PAZ E4, ranging between -14.1 ± 1.6 m MSL (Vlissegem: De Clerq et al., 2018; ID 353) and -4.1 ± 0.8 m MSL (Meetkerke), whereas terrestrial site Vossenhol (Heyse, 1979; ID 337) limits RSL to below -4.3 ± 0.2 m MSL during the same period. Offshore, borehole GR1 (De Clerq et al., 2018) reveals active tidal scour during PAZ E4, producing a marine lower limit at -38.3 m MSL. Lastly, in southwest Belgium, boreholes at Woumen and Kellen in the IJzer palaeovalley (Bogemans et al., 2016) provide SLIPs for the LIG highstand at ⁻5.7 m MSL (ID 346; including some decompaction and assuming 2 to 3.5 m tidal amplitude), and at -3.1 m MSL for an older interglacial (ID 347; MIS 7).



## 5.2 North Sea: Germany and Denmark

In NW Germany and SW Denmark, the LIG highstand shoreline ran roughly parallel to that of the modern Wadden Sea (Fig. 1), locally preserving tidal flat and supratidal marsh deposits below -7.5 m MSL. Where the shoreline turns north in the German Bight, it shows more estuarine indentations than in present times. These inshore settings, like in The Netherlands and Belgium, produced a series of 'highstand' tidal flat surfaces topped by regressive terrestrial limiting points. These are the German sites Leybucht and Land Hadeln (Streif, 1990; 2004: ID 880-884), Danish classic site Tønder (Madsen et al., 1909; Nordmann, 1928; Friborg, 1996; IDs 865, 866) and further sites Ribe, Esbjerg, Ringkobing and Harbooere (Konradi et al., 2005; IDs 869-873). The highstand appears to span PAZ E5 fully in NW Germany. For Danish-German sites where palynological control was absent, database entries encode it as equated to 'Cyprina Clay, upper saline phase' (Table 4), which in turn correlates to PAZ E4b-E6a (following Kristensen et al., 2000; Funder et al., 2002).

PaleoRSL for these SLIPs plots at -10 m MSL in NW Germany (Leybucht, Land Hadeln) but at least -5.5 m shallower in Danish-German borderland (Tønder, Ribe; marine limiting data only). Northward into Denmark, SLIP elevations fall to between -15 to -10 m MSL (Esbjerg, Ringkobing, Harbooere). A particularity is borehole site Vovov Bakkeoer (Konradi et al., 2005; ID 874) in the West-Danish offshore area, that revealed a patch of Saalian till/outwash deposits with Eemian marine cover, bearing foraminiferal evidence for later-shallower local submergence, than that of deeper waters surrounding it (e.g Horn Reef M3; ID 864). Hence, despite being positioned offshore the location produces a highstand SLIP (E4b-E6a) at -14.3 ± 1.3 m MSL. Terrestrial LIG exposures are obtained from just around modern MSL (-4 to +2 m) from shallow boreholes (Esbjerg, ID 871) and a classic exposures in low cliffs along the Danish Waddensea (Emmerlev Klev; Nordmann, 1928; ID 867). They place a terrestrial limit at -3.6 m MSL and at modern sea level (0 m MSL), to which inland limnic beds at Tønder add third such point at +2 m (Friborg, 1996; ID 866). Deploying VLM corrections (values in Fig. 4) to the German and Danish sites, places highstand elevations between -5.2 and +2 m MSL (and -7.1 m for Vovov Bakkeoer) and the terrestrial limit at +1.2 to +4.4 m MSL. The vertical uncertainty of these SLIPs ranges between ±0.5 and ±1.0 m and expands to a typical ±2.5 m when VLM is included.

A small second set of sites are those from boreholes in the Ems estuary and West-German Wadden Isles Borkum, Norderney and Spiekeroog (Streif, 1990, 1991, 2004; IDs 885-888). These cover a transgression SLIP (Borkum; ID 886, PAZ E3b) at -36 m MSL and a coeval terrestrial limiting at -16.5 m MSL (Ems Estuary, ID 885, PAZ E3), as well as terrestrial limiting points for two stages of regression, at -8.5 m MSL (Norderney, ID 887, PAZ E6) and -16.5 m MSL (Spierkeroog, ID 888, PAZ E6b.EW-Ia), coincident with the ending of the interglacial. With differential VLM correction (values in Fig. 4), the points plot at -28.6 and -9.1 m MSL (PAZ E3 transgression), respectively -8.5 and -16.5 m MSL (PAZ E6/EW regression). Uncertainties are ±0.5-1.0 m without and ±2.5 m with VLM correction.



A larger third set of sites, sampling from deeper buried estuarine and marine deposits filling outwash valleys inherited from the MIS-6 deglaciation, produces transgressive SLIPs, as well as marine limiting points for the highstand, and indications regarding Late Eemian regression. The lower parts of these valley fills (below -17 m MSL typically) survived erosion by glacio-fluvial outwash systems of Last Glacial age at sites Schnittlohe (Kosack and Lange, 1985; Lambeck et al., 2006; ID

889), Dagebuell (Winn and Erlenkeuser, 1995; Winn et al., 2000; IDs 875-877), Højer (Madsen et al., 1909; Nordmann, 1925; Konradi et al., 2005; ID 868) and Horns Reef M3 (Konradi et al., 2005: ID 864) in the German Bight, and sites Krummland (Winn and Erlenkeuser, 1995; Winn et al., 2000; ID 878, 879) and Tuschenbeck (Winn and Erlenkeuser, 1998; ID 890) along the Baltic Sea German coastline (Fig. 1). Inland-most coastal and estuarine beds are encountered in these boreholes. Especially along the SW Baltic settings allowing sea-level reconstruction are otherwise quite rare, as the Last

Glacial morainic limit has overrun and eroded superficial littoral LIG deposits.

Before marine inundation, most sites were limnic with imprecise water depth control, meaning that terrestrial limiting points could not be worked up. The moment of marine inundation is well established for these sites: latest at the inland position of Schnittlohe (PAZ E4), earlier in Baltic facing sites Tuschenbeck and Krummland (PAZ E3a). North Sea facing sites Horns

Reef M3 (basal peat), Dagebuell (beach facies at base) and Højer (shallow channel fill peat) are from valley floor rather than lake settings, which provide better opportunity to define RSLs. They register transgressive SLIPs at -40.0 m MSL (ID 864, PAZ E2a), at -24 m MSL (ID 875; PAZ E3a-E4b) and at -23 m MSL(ID 868, PAZ E4) and in Figure 5 plot as -33.8 ± 0.6 m, -20.3 ± 1.7 m and -19.3 ± 2.0 m MSL with differential VLM correction. From Dagebuell, Schnittlohe, Krummland and Tuschenbeck this is followed by marine limits to the highstand (PAZ 4b-E6a), the highest one at -12.5 m (Krummland; ID

879), -6.8 m (Dagebuell, ID 876) and -4 m (Schnittlohe, ID 889), corroborating and supporting the highstand SLIPs from the first discussed set of sites (also with VLM corrections). The top of the Eemian marine sequence at Dagebuell is intertidal, and hosts a regressive SLIP at -9.7 m MSL (ID 878; PAZ E6; -3.2 ± 1.9 m MSL with VLM correction), which is rare to have from N Germany (Fig. 5). Besides abundant palynological, foraminiferal, molluscan and high-density subsurface mapping providing control on setting, paleoenvironment and relative age for the above set of sites, from the Dagebuell core U-series

(base), oxygen isotopic investigations (middle), and an ESR date (top) are available, confirming MIS-5e ages (Winn et al., 2000).

Lastly, the NE of Denmark produces two deep-water marine sites, Skaerumhede I (Houmark-Nielsen, 1987; Larsen et al., 2009; ID 856) and Flakket, Anholt (Seidenkranz, 1993; IDs 857-861). The former is a deep research borehole recovering

some 80 meters of deep-water facies (Lower Skaerumhede Clay Fm.) deposited in the LIG equivalent of the Skagerrak channel (Fig. 1), with developed foraminifera and mollusca biostratigraphy (Knudsen and Lykke-Andersen 1982; Knudsen et al., 2009). It allows to tie water depths of 100-200 m to the facies encountered at -135 m MSL and translating in a single marine limiting point (ID 856) plotting at -17.5 ± 17.5 m MSL (Fig. 5). Site Flakket is also intensively marine palynological and biostratigraphically studied (Seidenkrantz, 1993), and has a water depth history of deepening from 35 ± 5 to 100 ± 10 m



water depth between the onset of the Eemian (ID 857; Kattegat Stadial; ID 858; PAZ E2a-E3a) and developed interglacial (ID 859; PAZ E3a-E4b; ID 860; PAZ E4b-E6a). This succession correlates with developments in the SW Baltic (Cyprina Clay phases, Table 4) and with the zonations of boreo-lusitanian molluscan and forma invasions and deepening trends of the Skaerumhede borehole. With water depth history accounted for, the marine limiting depths rise from -40 to -36.5 m MSL in the early Eemian, then accelerate to above +1 ± 15 m MSL (PAZ E3a-E3b) to stabilize above +22 ± 5.4 m MSL (highstand;

PAZ E4b-E6a). The top of the Flakket sequence shows a dropping water depth but the site stays marine during PAZ E6 defining a last marine limiting point (ID 861) at +13.0 ± 5.4 m MSL. The apparent superelevation of the LIG RSL data from these northwesternmost sites (Fig. 5) reflects the strong gradient in GIA fingerprint from this repeatedly ice-covered part of the study area, to the increasingly GIA-peripheral North Sea and Channel regions to the SW.

### 5.3 North Sea: Thames Estuary, East Anglia, North England

Sea level data points from the western North Sea coast primarily comprise of estuarine deposits from in or just above tidal frame. In the Thames, Preece (1999) describes a terrestrial limiting point (ID 3713) based upon the presence of freshwater molluscs throughout a sequence of organic silts from -2.4 to + 3 m OD, that overly a basal gravel (similar to the stratigraphy found at other locations in Trafalgar Square). The presence of the brackish water ostracod *Cyprideis torosa* and a few tests of the foraminifera *Elphidium articulatua,* which were likely carried upstream by tidal action, suggests this site was

deposited just beyond the limit of HAT. Similarly macro- (hippopotamus, hyaena and elephant) and micro-fossils (freshwater ostracods and molluscs) found in a channel sequence at East Mersea in Essex constrains a terrestrial limiting point (ID 3715) within the modern foreshore at ~3.2 m OD (Bridgland et al., 1995a; Briant et al., 2012). A further terrestrial limiting point (ID 4064) is present at Bobbitshole, Suffolk, which is the type-site for the Ipswichian interglacial, with the youngest LIG freshwater deposits as indicated by pollen and molluscs at approx. +1 m OD (West and Godwin, 1957; Sparks,

1957). The age of each of these terrestrial-limiting sites is constrained by state-of-the-art AAR dating of *Bithynia tentaculata* opercula, with ascription of a MIS 5e interglacial supported by the biostratigraphy (Penkman et al., 2013).

    The same AAR dating method (Penkman et al., 2013) is applied to the sequence from Tattershall Castle, Lincolnshire, where a salt marsh pollen sequence of detrital mud and organic silts contains grains of *Chenopodiaceae*, *Plantago maritima* and

*Artemisia*, alongside the brackish water ostracods and molluscs *Pseudamnicola confuse* and *Hydrobia ventrosa* (Holyoak and Preece, 1985). This allows us to ascribe this site to the salt marsh indicator type and a SLIP of -0.75 ± 0.71 m OD (ID 3736). Further north, on the modern East Yorkshire coast is the only non-estuarine MIS 5e deposit from this region; a chalk boulder beach (Catt and Penny, 1966; Lamplugh, 1887), overlain by windblown sands from which OSL samples were collected and dated to 120.9 ± 11.8 yr (Bateman and Catt, 1996). Due to the presence of landslide material covering the LIG

boulder beach (visited by NLMB in 2020) there is some uncertainty as to the exact elevation of the beach (ID 1380; 2.3 ± 2.8 m OD), which had to be estimated from the field diagrams (Bateman and Catt, 1996; Catt and Penny, 1966). Offshore of Norfolk two vibrocores recorded the LIG transgression as a sedimentary succession of sands and silts sat over a unit of

glaciofluvial gravel and gravely sand (Paddenberg et al., 2008, Russell and Tizzard, 2011). Silty deposits at ca. -29.54 m OD date to MIS 5 and display a micro-fossil assemblage dominated by the ostracod *Cyprideis torosa* and foraminifera *Ammonia*

*becarii*, *Haynesia germanica* and *Elphidium oceanis* indicative of intertidal environments (IDs 3432, 3433, 3434).

**Figure 6: SW-NE organised plot of Last Interglacial WALIS data entries for the English Channel, split to type and relative**

**position. Transect location in Fig. 1. Bottom panel shows sites and age-depth plot as a screenshot of WALIS' interactive viewer (Garzon and Rovere, 2021): https://warmcoasts.eu/world-atlas.html, which includes © OpenStreetMap contributors 2020 data as basemap (distributed under ODbL 1.0). Only LIG data points plotted.**





### 5.4 English Channel: Southern England

Sea-level data from Southern and Southwest Britain include a mix of marine and terrestrial limiting data, as well as precise sea-level index points. Several important sites along the Hampshire (Briant et al., 2006; 2019) and West Sussex (Bates et al., 2010) coastlines have been dated using OSL. These efforts have provided a welcome absolute chronology for the terraced sequences of raised littoral deposits along this stretch of the Southern Britain coastline. In West Sussex, the Aldingbourne Raised Beach sequence of marine sands and gravels found at elevations between +17.5 m and 27.4 m OD (at Norton Farm

and Pear Trea Knap; IDs 3716-3722) dates to MIS 7 based on replicate OSL sampling at multiple sites (Bates et al., 2010). The lower positioned Pagham Raised Beach sequence of marine sands and gravels, has been dated at multiple sites to MIS 5, and in places to MIS 5e (Bates et al., 2010). Its SLIP elevations are around +5 m OD (sites Selsey West Street, Pagham Water Treatment Plant, Chalcroft Nurseries, Warblington and Woodhorn Farm; IDs 3723-3733).

Fluvial deposits found at intermediate elevations between the Aldingbourne and Pagham Raised Beaches (Solent Breezes, IDs 3734, 3735) and dated to MIS 6, provide important terrestrial limiting data points (Bates et al., 2010). In Hampshire, several sequences have been dated that contain interglacial estuarine organic deposits located between lower and upper marine sand and gravel units (Allen et al., 1996; Briant et al., 2006; 2019). At Lepe, lower gravel units are OSL dated to MIS 6, classified as 'transgression' terrestrial limiting points (ID 3755, 3757, 3758). The upper gravel units date to MIS 4 (IDs

3753, 3754, 3756), placing the interbedded estuarine deposits in MIS 5 (SLIP IDs 3759, 3760), albeit that a paired date for the estuarine deposits is unavailable (Briant et al., 2006; 2019). These are to be regarded 'transgression' or 'regression' SLIPs because of Lepe's valley-central geographical position, and compared to other LIG SLIPs nearby relatively low elevations of -8.5 m and -2.6 m OD. Optical luminescence dates from overlying gravels in comparable sequences at Pennington Quarry (IDs 3761-3764) place buried estuarine deposits at -4.6 m OD in late MIS 5 (Allen et al., 1996; Briant et

al., 2006), and provide for 'regression' stage terrestrial limits. Buried interglacial estuarine deposits associated with the Solent River estuary system are further found on the Isle of Wight in association with the Bembridge Raised Beach (ID 4004; +4 m OD), which is dated with thermoluminescence to MIS 5e (Preece et al., 1990).

Raised interglacial beaches in both Devon and Cornwall (Scourse, 1996; Campbell et al., 1998) are commonly associated

within sequences of pebble conglomerate beach facies sitting on top of raised wave-cut platforms, and overlain by backshore sand facies that grade into aeolian dune sands. Four sites are included, all considered highstand/stillstand sites, of which only one (Fistral Beach, Newquay; ID 729) has a geochronology supported by thermoluminescence dating, independent from AAR (Southgate, 1985). That site dates overlying dune deposits (+9 m OD), providing a terrestrial limiting data point. The further three sites (Trebetherick, St Ives Bay, Saunton; IDs 528, 730, 735) each have AAR and stratigraphic information that

indicate MIS 5 provenance (Andrews et al., 1979; Bowen et al., 1985; Davies, 1983, 1984; James, 1968, 1975, 1995, 2008;



Gilbert, 1996; Scourse, 1996; Campbell et al., 1998), yet with low confidence in age quality. These plot as SLIPs at +4.2 to +7.2 m OD (Fig. 6).

The remaining MIS 5-associated data points included for Southern and Southwest Britain derive from raised littoral sediments found within cave systems in South Devon (Proctor and Smart 1991; ID 4008), South Wales (Sutcliffe and Currant 1984; Stringer et al., 1986; Sutcliffe et al.1987; IDs 4005-4007), and on Jersey (Keen et al., 1981; ID 4003). Each site contains speleothem deposits in association with the raised littoral sediments providing robust chronologies based on U-series dating. The Minchin Hole Cave (IDs 4005, 4006) site in South Wales is further supported with luminescence dates (Southgate 1985). Lastly, offshore in the English Channel, vibrocores dated by luminescence record a fluvial palaeosol (ID 3679) of late MIS 6 age at -42.9 m OD and laminated sand (ID 3678) rich in shell fragments and shallow marine foraminifera (*Elphidium* sp. and *Ammonia* sp.) of MIS 5 age at -48.8 m OD (Mellett et al., 2012; 2013).

## 5.5 English Channel: Northwest France

The most common sea level indicators in the French side of the English Channel are marine terraces and raised beaches. These features have been described from west Brittany to east Normandy as part of a staircase of coastal platforms with each step attributed to different interglacials (Coutard et al., 2006; Pedoja et al., 2011; 2014; 2018). The lower level is generally ascribed to the LIG and its age has been constrained by dating the deposits sat on top of the platforms (mainly heads, loess and beach sediments) or by the identification of Palaeolithic artefacts (Mousterian industry, Cliquet et al., 2003; 2009; 2013; Regnauld et al., 2003; Lautridou and Cliquet, 2006). Absolute ages are available at a limited number of locations and the rest of the sites are regionally correlated based on the developed stratigraphic frameworks (van Vliet-Lanoë et al., 2000; Bates et al., 2003; Regnauld et al., 2003; Monnier et al., 2011; Pedoja et al., 2018).

Marine terraces are generally found between 0 and +8 m NGF in mainland France (e.g., Trez Rouz, ID 3677; Cap La Hague, ID 3674; Asnelles, ID 3672) and in different islands distributed along Brittany's west coast (e.g., Belle Ile, ID 3667; Ouessant, ID 3658; Chausey, ID 3668). In Brittany, dune sands from deposits overlying an abrasion platform at Du Guesclin (ID 3659) were dated by Regnauld et al. (2003) providing a minimum age of 90 ka, and artefacts attributed to early Palaeolithic age were found overlying an equivalent platform at Le Verger (Regnauld et al., 2003; ID 3660). Absolute ages pointing to MIS 5 were also obtained from raised beach deposits at +8 m NGF at Piégu (ID 3657) by ESR on quartz grains (Bahain et al., 2012). Normandy shows the highest density of LIG sites, particularly around the Cotentin peninsula, including a suite of sea level index points, and marine and terrestrial limiting data. Sandy beach deposits that are found between 4 and 7.9 m NGF (IDs 3550, 3551, 3646, 3651, 3671) have been dated by OSL and TL to MIS 5, probably MIS 5e (Folz, 2000; van Vliet-Lanoë et al., 2006; Coutard et al., 2006; Cliquet et al., 2009), while aeolian sands (mainly dunes, IDs 3536, 3549, 3648) and marine deposits (ID 3552) provide ages for marine and terrestrial limiting points. Further east, in Le Havre, beach



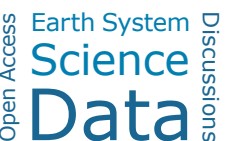

sediments found at -1 m NGF (ID 3676) has been interpreted as MIS 5e regressive deposits based on its stratigraphic position (Breton et al., 1991; Lautridou et al., 2003), but absolute ages are not available.


## 6    Chronological challenges

One of the challenges of any LIG sea-level database is the chronostratigraphic constraints, as developing chronologies for Pleistocene timeframes is outside the reach of radiocarbon dating. Even in low-latitude regions where absolute U-series dating can be applied to palaeo-corals, debate on the age constrains remains (e.g., Chutcharavan and Dutton, 2021). In our

study area, the MIS-6 glaciation limit divides the study area and the dataset quite evenly (N=68 within the  glaciation limit, N=73 outside).   For locations within the ice limit, the lower stratigraphic boundary left by the ice provides a strong maximum chronological constraint. In this area, the primary chronological challenge (section 6.1) is determining the absolute ages for the Eemian pollen biozones, as outlined in section 3.3 (Table 4). Outside of the ice margin, the limited lithostratigraphic controls means that assigning a pollen biozone to a specific one interglacial is a challenge, though recent

advances in AAR dating has led to some progress to resolve this problem (section 6.2). The chronological particularities and uncertainties of the North Sea region affect the basin subsidence corrections and graphic comparison of WALIS-visualised data against earlier work and syntheses (section 6.3).

### 6.1 The floating Eemian PAZs and their absolute chronology controversies

The North Sea PAZ entries in WALIS' chronostratigraphy relational tables (section 3.3, Table 4), bracket the floating E1-E6

scheme between two absolute numeric ages: 127 ka (acclaimed earliest possible start of PAZ E1) and 110 ka (latest possible ending of PAZ E6). This totals 17,000 years, which broadly equals estimates of the length of the LIG globally (Lisiecki and Raymo, 2005: 130-115 ka for MIS 5e), but is longer than the varve-counted 11,000 years of the scheme itself. Hence a user requiring absolute time control must decide where on the absolute time scale the Eemian starts, for which with current dating and correlation uncertainties, multiple scenarios exist (refs in section 3.3). The literature thus associates different numeric

ages to PAZ E1-E6. This spread in ages was taken over in WALIS, but with some filtering. The resulting suggested 'youngest' options may be more controversial than the 'oldest' options, and where the oldest option is put, may be debated on itself.

The youngest time option to consider is to have E1-E6 span 121 to 110 ka. In this option, the E5/E6 boundary is put at ~115

ka (Table 1), which would mean that climatic cooling and the start of the regression in the region correlate with global signals seen in Greenland ice-cores, deep-sea oxygen isotopes and coral-data dominated LIG sea-level history compilations (e.g. NEEM community members, 2013). The reason that the onset is as late as 121 ka, is that it would include the Blake palaeomagnetic event early on (Sier et al., 2015). Opting for this is controversial, as it implies that North Sea temperate



conditions started >5,000 years later than the onset of the interglacial conditions as considered in Southwestern Europe
(Portuguese shelf: Sanchez-Goñi et al., 1999; Shackleton, 2002; 2003: ~126 ka; Italian speleothems and Portuguese shelf:
Tzedakis et al., 2018: ~129 ka). It would imply that greater temperature gradients existed N-S across Europe than in the
Holocene. Explanations are sought in Scandinavia taking relatively longer to lose remaining ice mass (Kukla et al., 2002),
with repercussions for GIA and RSL (Long et al., 2015), both regionally and globally. One should only adopt the youngest
option, if one assesses scenarios where the Eemian course of events differed greatly from the Holocene.


Placing the start of the NW European Eemian at 124 ka is less controversial (this was the choice for Fig. 5 and 6). Placing
the onset at 127 ka, the oldest option, is not controversial. The time differences with the onset in Southwestern Europe then
are ~5 kyr (still some millennia a strong temperature gradient) and ~2 kyr (a 'near-simultaneous' onset) respectively. Such is
more in line with Holocene analogies, both in terms of the duration of the Scandinavian deglaciation and the timing of
regional GIA response to unloading. Investigating this further is an important foreseen application of the WALIS data, in
terms of the sea-levels signals (see section 7.1), and may also contribute to resolve the course of events of LIG regional
environmental and climatic change from South to Northern Europe.

An even older onset of the Eemian and hence the onset of sea-level rise in the North Sea, has been proposed by some
authors, notably: Funder et al. (2002): 132.5 ka and Beets et al. (2006): 131 ka. We filtered them (i.e. rejected them as oldest
possible ages), because these ages onsets so early appear now falsified by the Portuguese shelf studies that put the onset at
~126 ka, then pushed back to ~129 ka (Tzedakis et al., 2018), but not back to 131 or 132.5 ka. In the case of Funder et al.
(2002), the 132.5 ka age was based on a preliminary U-series chronology for midpoint Termination II that had ~135 ka as
age, to which their interglacial onset lags 2.5 kyr. Starting from the presently considered midpoint of Termination II, the
onset would be at 127.5 ka, coinciding with the 'suggested oldest option' (i.e. not controversial).

Lambeck et al. (2006) provides a GIA modeling study for Europe with focus on the MIS 6 to MIS 4 time period. It uses an
ice history with the Warthe substage maximum set at 143-140 ka and midpoint Termination II and onset Eemian as in
Funder et al. (2002). It reproduces the RSL curve of the Zagwijn (1983) data (Fig. 2a, Fig. 7) and from the cessecation of rise
(transgression) in the curve, deduces that 1.5-2.0 kyr further shifting is appropriate. Their 4.0-4.5 kyr after the midpoint
termination is 126-125.5 kyr on the current time scale. In this sense, the Lambeck et al. (2006) arrive results are in line with
the 127 kyr adviced 'oldest to consider' age.

What surfaces from the above summary of the age options and reasoning behind it, is that the different advocated ages for
onset of the Eemian (E1-E2), end of transgression / begin stillstand (E4b/E5), begin of regression and/or cooling (E5/6) and
onset of the Weichselian (E6/EW), all include dependencies to one or more ages from isotopic records, which in turn have
been subject to revision and choice. This is the nature of correlative means of establishing numeric ages, and the effect of





slowly increasing overall resolution and discovery of new records and techniques (U-series dating improvements). The sources from which tie-point ages were taken are (i) marine stacked records (e.g. the Termination II midpoint of Table 1; cf.

Lisiecki and Raymo, 2005 who derived it from Shackleton 2002, 2003), (ii) events in the Greenland ice-cores (e.g. Sirocko et al., 2005, for the ending of the interglacial), and (iii) ages established with U-series dating from higher resolution speleothems and transferred to single marine isotopic records (most recently: Tzedakis et al. 2018).

The revision of the interglacial onset age for Southern Europe to ~129 ka by Tzedakis et al. (2018) coincides with revision of

the midpoint Termination II in the same record to ~132 ka. The age should thus be seen as 3 kyr into the interglacial as recorded in main marine isotope signals, which was also the idea of putting ~127 ka as oldest onset, relative to the midpoint Termination II at ~130 ka, that is also the start age considered for MIS 5e used in WALIS during preparation of the dataset. This oldest' option has ~2 kyr lag time between onset interglacial in SW Europe at ~129 ka and the onset interglacial in In the North Sea part of our study area. The findings in Tzedakis et al. (2018) in Southern Europe may be reason to also push

Table 4's oldest ages back to ~129 kyr, as that would offer the the zero-lag time option to WALIS users too. Similarly, we should then also consider to update other elements in WALIS, i.e. the table storing peak, start and end age of the MISs ('form_MIS_ages'; Rovere et al., 2020), but the entry for MIS 5e is very widely used across the database and such changes could propagate.

## 6.2 Time control and data filtering in England and France

In the British Isles, biostratigraphy has also been typically used to characterise the sites into the pollen sub-stages of the 'Ipswichian' and the preceding 'Hoxnian' interglacials (West, 1977), but without the same lithostratigraphic constraints as in continental Europe, or the presence of a full Ipswichian pollen profile at a single site. It has since been shown that these 'pollen interglacials' have very similar vegetation profiles and in fact represent more than one interglacial (Thomas, 2001; Turner and West, 1968), with many early 'Ipswichian' sites conflated with those from MIS 7 (Lewis et al., 2011; Bridgland,

1994), and the 'Hoxnian' pollen spectra shown to be very similar in both MIS 9 and 11 (Roe et al., 2009; Thomas, 2001). Due to the limited litho-stratigraphic controls of UK Ipswichian sites, the lack of continuous a pollen profile (Lewis et al., 2011; Thomas, 2001), and the difference in the marine climate of the UK versus the relatively continental northwest European plain, it is not possible to simply correlate the 'Ipswichian' pollen zones with the Eemian zones from NW Europe (Turner, 2000). Though dating approaches have significantly improved since many of UK 'Ipswichian' sites were first

studied, not all sites are available to be revisited with modern methods; and therefore, several sites which some may consider to be evidence for LIG sea level have been discounted from our database e.g Somersham, Cambridgeshire (West et al., 1999), Kirmington, Lincolnshire (Straw, 2018), Burtle Beds, Somerset (Kidson and Heyworth, 1976), plus 6 of the 7 Thames sites in Hollin's (1977) analysis (section 2.5). Furthermore, questions around microfossil and sediment reworking/preservation e.g. Tattershall Thorp, Lincolnshire (Holyoak and Preece, 1985) and Somersham (West et al., 1999),

and debate around the presence/absence of fauna and flora during specific interglacials (e.g., Meijer and Preece, 2000) means



sites that cannot conclusively be ascribed to a single interglacial period, or indicative meaning, are excluded from the database.

To address some of the issues in biostratigraphic chronology across Britain, relative dating of interglacial coastal sites using amino-acid racemisation (AAR) began in 1979 (Andrews et al., 1979; Miller et al., 1979) and continued apace throughout the 1980s (Keen et al., 1981; Campbell et al., 1982; Davies 1983; Andrews et al., 1984; Bowen et al., 1985; Davies and Keen 1985). Such investigation was also performed on material from multiple marine interglacial sites collected from across Europe (Miller and Mangerud, 1986). A landmark review of these early works is given in Penkman et al. (2010). Early AAR results often indicated potential associations with two or more interglacial periods within individual sites, or even multiple ages within single interglacials. Major challenges were the evolving preparation procedures and inconsistent choices of species for sampling, which both hindered comparability of AAR values across different studies. Around the same time, results from the application of novel dating techniques to raised interglacial deposits in the region were being used to inform the AAR debate. U-series dates from the raised beaches at Belle Hogue Cave on Jersey (Keen et al., 1981) and at Minchin Hole (Sutcliffe and Currant 1984) and Bacon Hole (Stringer et al., 1986) in South Wales were often cited to correlate AAR-dated deposits to MIS 5e (Davies 1983; Bowen et al., 1985; Davies and Keen 1985; Bridgland et al., 1995b), although single U-series dates were insufficient for describing the full complexity of single sites given the multiple (age) groupings that were often present within AAR analyses. Early thermoluminescence techniques were also being used at this time (Southgate 1985) but were still unable to distinguish interglacial associations as results from fine grained and large grained sand fractions provided dates of MIS 5e or MIS 7, respectively.

Improved certainty in the AAR debate was only reached in 2013 when an aminostratigraphy for the British Quaternary was developed using the opercula or the gastropod species *Bithynia tentaculata* (Penkman et al., 2013). Unfortunately, none of the existing AAR results from interglacial coastal sites in southern and southwest Britain were based on this species, and material from only limited sites in eastern England is available, and therefore definitive age determinations for many of these sites remain elusive, resulting in some sites being excluded from the database. Subsequent development and application of luminescence and U-series dating techniques has gone some way to confirming interglacial associations at several sites (Preece et al., 1990; Proctor and Smart, 1991; Bates et al 2010; Briant et al., 2019), though there are numerous sites in both the UK and France that lack independent dating control (e.g., Tastet, 1999, Haslett and Curr, 2001). With a few exceptions, most of the sites in northern France have been attributed to the MIS 5 based on absolute ages that were obtained from deposits overlying or underlaying the LIG features, and in most cases these results are not enough to discard older or younger stages as the origin, There remains considerable opportunities to revisit many (known and as yet undescribed) interglacial raised beach sites across southern and southwest Britain and France and to apply now established sea-level research protocols (Shennan et al., 2015) and spatially intensive geochronology sampling strategies that could combine with statistical (e.g. Bayesian) frameworks to greatly advance knowledge and data precision across this region.



## 7 Closing Remarks

### 7.1 Near Field importance in resolving global Last Interglacial sea levels

The main foreseen usage of this database for the North Sea and Channel as Near Field region of the European ice sheets, is in GIA modelling and analysis of RSL fingerprints to appropriately attribute these to southern and northern hemisphere respective sources. The pattern and magnitude of RSL change at near field sites, proximal to existing or former ice sheets, are strongly dependent on the magnitude and timing of the change in ice load and the relaxation characteristics of the underlying mantle (Farrell and Clark, 1976; Yokoyama and Purcell, 2021). Therefore, near field sea level observations have the power resolve ice sheet histories and changes in the solid Earth that far-field data do not; as has been extensively demonstrated with near and intermediate field Holocene RSL data (e.g., Lambeck, 1995; Peltier, 2002; Lambeck et al., 2006; Bradley et al., 2011; Engelhart et al., 2011; Long et al., 2011). The relative density of data in NW Europe, over a transect away from the Saalian ice sheet margin (Figs 5 and 6), means RSL data from this region has the potential to reconstruct the magnitude and deglaciation of the MIS 6 Eurasian ice sheet, which is currently poorly constrained and important for both near and far-field LIG sea level (Rohling et al., 2017; Dendy et al., 2017). Near-field sites can also take advantage of the sea-level fingerprint of ice sheet mass balance changes to constrain the source of ice sheet melt (Tamisiea et al., 2001; Mitrovica et al., 2009). Ongoing debate suggests asynchronicity between the timing of the contribution of the Greenland and Antarctic ice mass loss to LIG barystatic sea level (Rohling et al., 2019; Turney et al., 2020); a hypothesis which near-field RSL data can test (Kopp et al., 2009; Hay et al., 2014; Long et al., 2015).

One of the challenges of near field RSL data is that regional GIA is a dominant component of the overall sea-level signal. This does in turn present advantages, as alongside long term VLM, GIA can provide the accommodation space for the accumulation and preservation of (near-)continuous late Quaternary sedimentary packages (e.g. Eaton et al., 2020). Interglacial RSL highstands will occur earlier in the far field than the near field due to solid Earth processes, with the initial transgressive phase being relatively slow and therefore having the potential to capture fluctuations in RSL (Cohen et al., 2012; Long et al., 2015). Then at the end of the interglacial, RSL fall in the near-field is relatively rapid, as changes in ocean volume due to growth of ice sheets of the commencing glacial phase, outpace regional GIA. Near-field sites often include temperature and high-latitude estuarine sequences (e.g. salt marsh) which due to their close relationship with tidal levels has the potential to provide accurate and precise vertical constraints on RSL (Shennan et al., 2015); with developments in chronological techniques only enhancing their potential to deliver 5* LIG SLIPs. This needs to be an area of intensive research focus, as near-field RSL constraints from pervious warm periods are essential to identify the sources and forcing mechanisms responsible for sea-level change (Dutton et al., 2015).

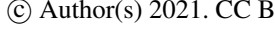



## 7.2 Comparison to older interglacials

For older interglacials, the current version of the NW Europe database contains a modest subset (Fig. 1): 5 SLIP sites along the North Sea (Morton, Norfolk UK; Sangatte; NW France, Kellen, Belgium; Noorderhoeve and Ameland in The Netherlands; IDs 4063, 3665, 347, 895, 429), and along the Channel the OSL-dated Aldingbourne raised beach (West Sussex, UK – see 5.4). All of these are attributed to MIS 7 (Gale et al., 1988; Hoare, 2009; Balescu et al., 1997; Bogemans et al., 2016; Meijer et al., 2021) and all of them are from highstand positions. As for the LIG highstand, the indicators occur in raised position along English Channel, East Anglia and also in the Dover Strait (Sangatte). They occur in subsided position in The Netherlands (VLM rates included in database and Fig. 4). The series of MIS-7 attributed sites along the Thames Estuary (section 6.1) has not been entered. Our compilation efforts focused on the LIG. We do recommend to expand WALIS' contents also with such entries for older interglacials.

The Belgian site is from the IJzer-valley and in facies and setting similar to the LIG counterpart Woumen (Bogemans et al., 2016; ID 346). The Morton site is a beach deposit with no clear nearby counterpart in East Anglia. Sangatte is also without preserved LIG counterpart, attributed to Holocene highstand erosion (section 5.5). The Dutch sites (Meijer, 2002; Meijer et al. 2021; IDs 429, 895), are from estuaries overridden by the MIS-6 glaciation. In terms of depositional environments, they are similar to LIG and Holocene settings, but in terms of geographic position and orientation they are dissimilar. Geological Survey investigations in the Northern Netherlands (Bosch, 1990; and later and ongoing work) reveals a complex of glaciogenic (older than MIS 6), estuarine and fluvial deposits, including regionally traceable peat and shallow intertidal levels, similar in degree of preservation as such from the LIG. This will certainly provide opportunity to add older-interglacial RSL data points from this region in the near future contributing to insights in sea-level position during the Middle Pleistocene, potentially not solely 'highstand' but also 'transgression' data points.

## 7.3 Comparison to Holocene sea level indicators

For reasons of brevity and focus, we only loosely referred to Holocene record and do not attempt to provide an overview. There is a long history of research into Holocene RSL sea-level indicators from this region, in particular in the UK and Netherlands (Shennan, 1989; Flemming, 1982; van de Plassche, 1982), the databases of which provide the foundation for the design of modern RSL databases (Hijma et al., 2015). For recent datasets, we refer to Vink et al., (2007) and to HOLSEA regional publications (Shennan et al., 2018; Hijma and Cohen, 2019; Bungenstock et al., 2018). It should be realised, however, that analogies drawn between Holocene and the LIG are implicit part of the data review and reformatting exercise for this paper and the WALIS database, in particular the indicative meaning (Shennan, 1986; van de Plassche, 1986).



### 7.4 Future data collection directions

In this data paper, we separated transgression, highstand/stillstand and regressive groups for graphic presentation and to guide the provided overview. The last part of the transgression, the highstand, and the very beginning of regression is fairly covered and internally replicated, by over 50 datapoints from the area, with some sites even providing 2-3 successive SLIPs. In comparison, the earlier rising limb (first half MIS 5e) is not yet well covered, and neither is the falling stage (MIS 5 as a whole, and continuing in MIS 4 and 3).

The transgressive part of the data set (Table 1), potentially includes record of acceleration and deceleration phases in near-field regional RSL, that in far-field RSL records are more cryptically or not recorded. Even with VLM corrections deployed, the mean uncertainty of individual transgressive SLIPS (N=19) is ±2.5 m in the NW-European database and is ±1 m for the best sites. At present, the set of transgressive slips, especially those from the earlier part of the transgression, suffers from

1245 being spatially distributed and each site needing its own VLM corrections (that are uncertain, section 4.3) to resolve the transgression signal and quantify contemporary rates of RSL. In this light, targeted collection of a series of transgressive SLIPS from along submerged Saalian palaeorelief off the Dutch coast (setting in Cartelle et al., 2021) has been caried out (ERC RISeR project; Barlow cs.), and are anticipated to append the current dataset.

The offshore 'regression' data points in the data set (Fig. 1), likely include signal from RSL oscillations and (sub)highstands of MIS5c and MIS 5a (Table 1: Falling stage), and potentially even from within MIS-3. The number of datapoints is sparse for this timeframe, however, and a data collection effort with different strategy than for the LIG highstand is required to resolve the signal. The offshore record potential for the Late Pleistocene is not restricted to terrestrial limiting data from fluvial settings alone, but also includes marshy deltaic and shallow marine strata. Potential appears particularly large in the

Southern Bight, where new generations of seismic surveying instruments and analysis capacity are used for targeted vibro-coring of submerged landscape features (e.g. Missiaen et al., 2020), including at places with sea-level indicative potential. In this area and region, *falling stage* sea-level research interest links up with that for Neanderthal and Mesolithic archaeology (e.g. Hijma et al., 2012), and with intensifying human activity in the near shore for windfarm construction and dredging for coastal nourishment and seaward harbour extensions (e.g. Cartelle et al., 2021). Such adds to demand for insight into

regional RSL history of MIS 5 (that the WALIS database may cater), and to data collection potential from which research may profit. Lastly, the potential for MIS 5c-a RSL datapoints is not restricted to just offshore, but extends to the North Sea coastal zone and the western rim of Holland, where basal parts of younger than LIG estuarine and fluvial-tidal channels appear preserved below younger Rhine and Rhine-Meuse deposits (Törnqvist et al., 2000; Wallinga et al., 2004; Busschers et al., 2005; 2007; Hijma et al., 2012; Peeters et al., 2016), with OSL-dating the most readily deployable dating technique in

that setting, as also offshore (e.g. Mellett et al., 2012; 2013; De Clercq et al., 2019).

On land, new data collection in general need not be constrained to finding new sites. Many sites from older literature with some effort are still possible to revisit, collect new data from and upgrade the quality of the chronologies. This has been carried out with focus on older interglacials in southern and eastern England (iGlass project NE/I008675/1; Long et al., 2015; Barlow et al., 2017) and the three (!) rounds of re-coring and resampling for Dutch site Amersfoort (section 2.1) and 1990ies work at Tønder (Friborg, 1996) and Dagebuell (Winn et al., 2000) can also be classified as scientifically successful revisits. Importantly, choices of what legacy site to visit could be more geographically inspired (filling gaps), instead of returning once again to classic sites that are known to be quite good already.

## 8    Data availability

The NW Europe database (Cohen et al., 2021; https://doi.org/10.5281/zenodo.5608459), as a scientific product, is open access. The data points used in this study were compiled and contributed to WALIS by the authors (see section 2) and in various ways the entries cross-refer to (i) governmental databases (with public portals, but their contents not open data / open access in the academic output sense) and (ii) to tabulated and graphed data contained in recent and legacy literature (in great majority web disclosed, not in all cases open access, DOI referenced where appropriate). The files at this link were exported from the WALIS database interface on 27 October 2021. A description of each field in the database is contained at https://doi.org/10.5281/zenodo.4459297 (Rovere et al., 2020), readily accessible and searchable at https://walis-help.readthedocs.io/en/latest/ More information on the World Atlas of Last Interglacial Shorelines can be found at https://warmcoasts.eu/world-atlas.html. Users of our database are encouraged to cite the original sources alongside with our database and this article.

## Author contributions

KMC, RB, VC and NLMB each reviewed regional bodies of literature (see section 2), compiled the data, assigned indicative meanings, and documented WALIS database entries that underpin this paper. KMC and VC prepared figures and tables. FSB and KMC assessed North Sea basin subsidence VLM. All authors edited the manuscript (in the template of the WALIS database special issue), designed figure and table legends, and addressed referee comments.

## Competing interests

The authors declare that they have no conflict of interest.

## Special issue statement

This article is part of the special issue "WALIS – the World Atlas of Last Interglacial Shorelines". It is not associated with a conference.



**Acknowledgements**

This paper forms a contribution the RISeR project, which has received funding from the European Research Council (ERC) under the European Union's Horizon 2020 research and innovation programme (grant agreement no. 802281; PI Barlow). The authors acknowledge PALSEA, a working group of the International Union for Quaternary Sciences (INQUA) and Past
Global Changes (PAGES), which in turn received support from the Swiss Academy of Sciences and the Chinese Academy of Sciences. We are grateful to Alessio Rovere (WALIS lead and special issue editor) for guidance and assistance in the preparation of this paper, Kirsty Penkman (University of York) for discussion of the AAR data compilation, Sebastian Garzon (internship student from University of Münster) for developing the WALIS visualization interface and Amy McGuire (University of Leeds) and Chronis Tzedakis (University College London) for discussion around UK interglacial
pollen. The data used in this study were compiled in WALIS, a sea-level database interface developed by the ERC Starting Grant WARMCOASTS (ERC-StG-802414), in collaboration with the PALSEA (PAGES–INQUA) working group. The database structure was designed by Alessio Rovere, Deirdre Ryan, Thomas Lorscheid, Andrea Dutton, Peter Chutcharavan, Dominik Brill, Nathan Jankowski, Daniela Mueller, Melanie Bartz, Evan Gowan and Kim Cohen. Reviews and comments by (TO FOLLOW) greatly helped in preparation of the final paper.

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
