# Peer review of "Last Interglacial sea-level data points from Northwest Europe"

_Earth System Science Data, 2021_

## Referee Comment (RC1)

**Review of *ESSD*-manuscript "Last Interglacial sea-level data points from Northwest Europe" (Cohen et al.)**

Cohen et al. synthesize sea-level data from the last interglacial (MIS 5e) in NW Europe by means of the standardized and publicly available WALIS database. Despite the long history of relevant research in this region, to the best of my knowledge this is the first systematic compilation of LIG sea-level data from NW Europe. As the authors point out, the ice-marginal setting of the study area makes it an interesting complement to the more established LIG datasets from far-field regions. Clearly, the authors have made an effort to capture the full literature on this subject, stretching back into the 1800s, which I appreciate. However, the presentation needs a lot of work. My review focuses mainly on conceptual issues and less on regional specifics (Sections 2 and 5). I also am not an expert on last interglacial chronology (Section 6), which is an important issue given the paucity of numerical ages and the critical role of stratigraphic correlation/interpretation.

My main concerns revolve around the following issues: (1) an apparent *a priori* interpretation of the RSL history; (2) terminology, definitions, and illustrations; and (3) the length of the manuscript.

[1] The authors quite deliberately work toward a transgression/highstand/regression interpretation of LIG RSL change that culminates in Section 5. I have several concerns here. First off, and fundamentally, these terms and their sequence-stratigraphic connotation (lines 764-765) concern shoreline change as a function of RSL change **AND** sediment supply. In other words, the link with RSL is not necessarily straightforward. My larger concern is that such an interpretation reeks of preconceived notions, i.e., that one single RSL cycle (i.e., rise, highstand, fall) occurred during the LIG. As they are well aware (e.g., Barlow et al., 2018, *NG*; somewhat curiously not cited in this manuscript), alternative views with multiple m-scale LIG RSL cycles have been advanced in the literature (e.g., O'Leary et al., 2013). I am not saying that I necessarily buy into these alternative views and/or that the authors are incorrect. However, using this assumption at the outset flies in the face of the goal of a sea-level database: to synthesize data as objectively as possible and to leave the door open for testing competing hypotheses. This is especially true for a dataset like the present one, with relatively large chronological and vertical uncertainties. On a somewhat related note, in lines 1238-1239 the authors refer to MIS 5 as a whole as "falling stage." This is a gross oversimplification, given the major subsequent highstands (MIS 5c and 5a) as documented from many localities worldwide (and note the contradiction with lines 1250-1251). In conclusion, I think there are multiple reasons to back away from the transgression/highstand/regression model as presently used.

[2] Section 3 features a discussion of sea-level indicators, including a few that are newly introduced. This is where accurate and consistent terminology, along with illustration, is critical. So why not include a cartoon with a schematic cross section and sedimentary logs that illustrate the various sea-level indicators, their stratigraphic position, and their

interpretation (SLIPs versus limiting data)? Quite frankly, I would consider that more important than some of the panels in Figs. 2 and 3.

On key terminology, is Relative Water Level something different than the widely used Reference Water Level? Also, with respect to the RWL and IR calculations it would be helpful to clarify why MSL is used, as opposed to the commonly used MTL (e.g., Shennan, 1986). I partly bring these things up because I was struck by the fact that the indicative range for most of the sea-level indicators is the same, which raises the question to what extent they are mutually exclusive.

With regard to basal peat, it is fine to initially interpret this as a GWL indicator, but (1) it is unclear what the depth range information in line 497 is based on; and (2) by not including the next step (i.e., how basal peats can become SLIPs) it isn't clear how these could be anything more than limiting data. The main element that is missing here is paleoenvironmental information (e.g., biological, geochemical) that is necessary to determine whether basal peat is intertidal. From this perspective, it is unfortunate that biological information ("marine fauna") is divorced from the other sea-level indicators. Absent this type of evidence, an interpretation where the initial stage of basal peat formation occurred in a fresh environment (lines 489-491) becomes rather speculative. Separate from this, note that basal peat is commonly defined as resting immediately on a largely compaction-free substrate, so other than thickness loss of the sampled interval, there would be no vertical displacement. For tops of peat beds this is of course different but referring to those as basal peats (lines 671-672) is bound to lead to confusion.

I was unfamiliar with the concept of estuarine terraces (Section 3.2.4). Elsewhere (line 672) I came across the term "estuarine tidal flat tops" and the spreadsheet mentions "preserved tidal flat surface." What is needed here, is a clear explanation of what this would constitute in a modern tidal/estuarine environment. Were these tidal flats? And if so, how exactly is it defined? Note that it is not uncommon to restrict tidal flats to mudflats that reside between MTL and MLW (or MLLW), i.e., excluding marsh platforms (e.g., Fagherazzi et al., 2006, *PNAS*). The formulas indicate otherwise, however, placing this category in the same elevation range as coastal marshes... so what is the difference? As mentioned before, a cartoon would be of great help here – it would enable the authors to more effectively communicate the relationships between the different sea-level indicators. As mentioned, the different categories must be mutually exclusive, but are they? Along similar lines, is there a difference between marine terraces and abrasion platforms (lines 1032 and 1034)?

With regard to elevation (Table 5 and associated text), cross sections are not a measurement technique, merely a means to illustrate data. What matters here is how elevations as shown in a cross section were determined. Also, what does "±10% of elevation measurement" mean? For example, if an elevation of 10 cm above MTL is obtained, the error would be 1 cm, but if it is 1 m, the error would be 10 cm? I doubt that this is what the authors meant. A larger question is whether elevations are related to modern MTL or simply to a geodetic datum. If the former, how do the authors account for the fact that MSL is subject to fairly rapid change, often of opposite sign as in Britain

(lines 642-645)? Finally, vertical uncertainties of only ±1 m (line 1243) for the LIG seem unrealistically low. That would place them among the best RSL data even within much of the Holocene. I find it difficult to see how adding >100 kyr would not inevitably result in larger errors. Am I missing something?

[3] Portions of the text (notably Sections 2 and 5) tend to get lost in lengthy regional stratigraphic details, with associated nomenclature that many if not most readers will be unfamiliar with. For example, the authors repeatedly mention the Lusitanian (a chronostratigraphic unit?) that I couldn't find in Table 1. Lincolnshire is also mentioned multiple times but not shown on a map. Note that these are merely examples. Trimming down this section would make the paper much more accessible. Would it be possible to move most of the regionally specific text to a supplement?

I came across tons of grammatical errors. The nature of the work is such that accurate communication is essential. I know that this is more about bookkeeping than imagination (and thus not particularly exciting) but failing to do so might cause more confusion than resolution. Put differently, other authors should be able to reproduce the work. In its current form, the writing is quite rough, and at many occasions difficult to follow. For a small selection of examples, see lines 147, 213, 300-301, 334, 398-399, 435, 625, 727-728, and 1091. Without a doubt, an effort by all authors can address this, not least the native English speakers on the team.

Some more specific comments regarding the text:

Line 55: I suggest adding Shennan (1982, *Proc Geol Assoc*) here. Also, if the journal guidelines allow it, I would recommend listing references chronologically.
Line 80: there is a widespread preference for "numerical ages" over "absolute ages" (see Colman et al., 1987, *QR*).
Line 99: not sure what "superregional" means.
Line 104: you probably mean "paleoceanographic" here.
Line 115: why not just stick to "sea-level data points" as in the title?
Line 125: better to say "unlike the present interglacial."
Line 143: I suppose this must be cm?
Lines 194-195: not clear what a "sea-level site" is.
Line 290: you define Cyprina Clay here, but it has already been mentioned earlier (line 239) without a definition.
Line 308: here and elsewhere (wherever appropriate) I suggest you use "succession" rather than "sequence" unless it explicitly refers to sequence stratigraphy.
Line 313: please try to be more precise with terminology. Note that littoral environments occur in lakes as well, and lacustrine is the more commonly used term rather than limnic. Why not just referring to this as lacustrine versus marine (or shallow marine)?
Line 447: here and elsewhere, the authors use the term "generic" (or "non-generic") but it isn't particularly clear what this means.
Line 451: "subaquatic" should be "subaqueous."
Lines 461-465: what is said here about tidal amplification/dissipation matters a great deal for sea-level reconstruction and needs to be supported by one or more references.

Line 481: the definition that is provided here is debatable; basal peats typically overlie the transgressive surface that represents the upper boundary of the lowstand (or falling stage) systems tract (e.g., Zaitlin et al., 1994, *SEPM Spec Pub*).

Lines 485-487: the paleosol interpretation is tenuous, especially in the absence of references. The evidence I'm familiar with (e.g., Vetter et al., 2017, *G3*) shows that paleosols formed immediately prior to basal peat accumulation. Or are the authors referring to entirely different paleosols? Please clarify.

Line 571: the RWL should be defined more precisely; I presume it is this range divided by two?

Line 597: "footer" should be "footnote."

Line 620: is the next section really about the English Channel? It certainly doesn't appear to be about chronostratigraphy?

Lines 634-636: are you sure "subtract" is correct here? Shouldn't this number be added to bring an elevation from low tide level to mean tide level?

Lines 665-671: Keogh et al. (2021, *JGR-ES*) discuss the time-dependent nature of compaction of organic-rich coastal deposits, including its implications for sea-level reconstruction.

Line 678: is the nature and thickness of overburden included in the database?

Line 696: note that VLM strictly includes compaction, but that is clearly not how the authors define things in this work. This needs to be clarified somewhere.

Lines 701-703: I have trouble following this sentence; what does "earlier applied basin subsidence" mean?

Lines 731-733: using uncertainties is good, but it raises the question why no uncertainties are used for areas outside of the North Sea Basin where VLM appears to be less well understood (also see my earlier comment about vertical uncertainties). This must be addressed.

Line 1202: expressions like "5* LIG SLIPs" are not really appropriate for a paper like this one. If there is a ranking of data point quality, this should be clarified.

Lines 1231-1233: the authors seemingly intend to make a significant point here, but I had trouble understanding what that point is. Are they highlighting the differences or the similarities between WALIS and Holocene RSL databases?

Line 1281: this link took me to a southern African database. I suppose something else was intended here?

The quality of the figures needs work; for example, the lower portions of Figs. 5 and 6 are very difficult to read. A few more specific suggestions regarding figures and tables:

Fig. 1: the differentiation between small and large rivers is not meaningful (I can't tell the difference in the map). The depocenter VLM information from Fig. 4 shouldn't be repeated here. Instead, make sure that all geographic names mentioned in the text can be found in the maps.

Fig. 2 is not of the greatest quality and is every panel referred to in the text?

Fig. 3: please show the location of these areas in Fig. 1.

Table 1: Last Interglacial covers all of MIS 5, which seems inconsistent with the LIG definition used throughout the paper (MIS 5e or Eemian; also see line 100).

Table 2 needs a lot of attention; it contains a ton of text, and numerous acronyms that are not explained (note that it should be possible to read the table content without having to consult the main text). It would probably help to simplify things a bit. For example, could a finite list of geomorphic/stratigraphic contexts be provided and show which ones occur in a given geographic region? Maybe the table is simply too large, and this aspect should be separated from the numbers of data points in each geographic region.

Table 3: within this table there is a header that says "Paper Author added as part of this study." I had to read this several times, but I'm still not sure what it means. Also, please specify which other WALIS Special Issue papers are being referenced here.

Table 4: the title is awkward; please reword. And what exactly is meant with "Duration constraint"? Is this the maximum duration? I looked at the various numbers in the table but couldn't make sense of this.

Torbjörn Törnqvist

---

## Author Response (AR1)

**Author Comment response (AC1)** **+ full list of revision actions in response to RC1 and RC2**
**Preprint manuscript and Discussion: https://essd.copernicus.org/preprints/essd-2021-390/**

We thank the two reviewers for their comments and their patience with our manuscript.

We apologize for the many inconsistencies in the original preprint, that was produced under pressure. We appreciate that the reviewers recognized the relevance, effort to capture full literature and geographical qualities.

We have used the comments to majorly edit and rearrange the text to provide a revised manuscript, that we will upload shortly. The revision has included moving some sections to an appendix. As part of this we have also developed version 2 of the WALIS database (amended online, and an export version uploaded as V2 in the Zenodo entry) with a few additional data points and clarifications.

*Kim Cohen, Natasha Barlow, on behalf of all authors -- 29 April 2022*

**Here we respond to the RC1 and RC2 comments:**

**Reviewer Comment 1 (RC1) – (1/3):**

Cohen et al. synthesize sea-level data from the last interglacial (MIS 5e) in NW Europe by means of the standardized and publicly available WALIS database. Despite the long history of relevant research in this region, to the best of my knowledge this is the first systematic compilation of LIG sea-level data from NW Europe. As the authors point out, the ice-marginal setting of the study area makes it an interesting complement to the more established LIG datasets from far-field regions. Clearly, the authors have made an effort to capture the full literature on this subject, stretching back into the 1800s, which I appreciate. However, the presentation needs a lot of work. My review focuses mainly on conceptual issues and less on regional specifics (Sections 2 and 5). I also am not an expert on last interglacial chronology (Section 6), which is an important issue given the paucity of numerical ages and the critical role of stratigraphic correlation/interpretation.

My main concerns revolve around the following issues: (1) an apparent a priori interpretation of the RSL history; (2) terminology, definitions, and illustrations; and (3) the length of the manuscript.

[1] The authors quite deliberately work toward a transgression/highstand/regression interpretation of LIG RSL change that culminates in Section 5. I have several concerns here. First off, and fundamentally, these terms and their sequence-stratigraphic connotation (lines 764-765) concern shoreline change as a function of RSL change AND sediment supply. In other words, the link with RSL is not necessarily straightforward. My larger concern is that such an interpretation reeks of preconceived notions, i.e., that one single RSL cycle (i.e., rise, highstand, fall) occurred during the LIG. As they are well aware (e.g., Barlow et al., 2018, NG; somewhat curiously not cited in this manuscript), alternative views with multiple m-scale LIG RSL cycles have been advanced in the literature (e.g., O'Leary et al., 2013). I am not saying that I necessarily buy into these alternative views and/or that the authors are incorrect. However, using this assumption at the outset flies in the face of the goal of a sea-level database: to synthesize data as objectively as possible and to leave the door open for testing competing hypotheses. This is especially true for a dataset like the present one, with relatively large chronological and vertical uncertainties. On a somewhat related note, in lines 1238-1239 the authors refer to MIS 5 as a whole as "falling stage." This is a gross oversimplification, given the major subsequent highstands (MIS 5c and 5a) as documented from many localities worldwide (and note the contradiction with lines 1250-1251). In conclusion, I think there are multiple reasons to back away from the transgression/highstand/regression model as presently used.

*Reply (AC1):*

Thank you for the comments regarding our previous use of these terms. This has triggered major revision of the presentation, to fix some matters that could be misunderstood from the manuscript.

Although the previous version gave the review the impression that we were 'working towards a transgression/highstand/regression interpretation of LIG RSL change', we can reassure this was not the case. This confusion arose from the fact that the North Sea part of the study area (Fig. 5) provides SLIP data from early, middle and late stages of the interglacial that come from chronostratigraphically and geographically separated positions in the overall fairly continuous basin fill. In the source papers, the SLIPs have been picked from high resolution documented subsurface deposits, that prior to deciding what observation would be a SLIP (or TL, or ML) had been collected as part of local mapping. That local mapping had involved lateral and vertical tracing and palaeoenvironmental characterization of various strata. This mapping included identifying submergence (vertically, sedimentologically and biostratigraphically) and transgression (from lateral tracing facies representing paleoenvironments) etc. In other words, original authors, and also we in our WALIS cataloguing efforts, worked *from* findings of 'transgression' in the classical sequence stratigraphy sense, and not *towards* interpretation of transgression in the RSL sense. Our cataloging effort has mainly been on properly separating legacy sea-level points into those which are truly SLIPs, from those which are better regarded a Terr. (Upper) Limiting or Marine (Lower) Limiting points and the assessment of the vertical position (see comment 2). If the impression was that we upscaled local to regional findings (country by country along the North Sea) to a global signal we regret this and regard it due to poor presentation for which we apologise. The paper intends to be simply a summary of data from the region as requested for the WALIS special issue: listing the classifications and methods, summarizing the cloud of data and stopping prior to interpretation as to the patterns and drivers of RSL.

To address the reviewers concerns the manuscript now strongly reduces use of the terms transgression, highstand, regression throughout, making much clearer that we simply use these terms to describe the sedimentary sequences, rather than making inferences about the patterns of regional and global sea level. Similarly, we have removed these terms in the presentation of data in figure 5 and 6 and reclassified the data points (where chronological information allows) to early, middle and late LIG. However, we must stress this is within the context of the regional relative chronology and not global assessment of LIG climate/duration.

**Reviewer Comment 1 (RC1) continued (2/3):**

[2] Section 3 features a discussion of sea-level indicators, including a few that are newly introduced. This is where accurate and consistent terminology, along with illustration, is critical. So why not include a cartoon with a schematic cross section and sedimentary logs that illustrate the various sea-level indicators, their stratigraphic position, and their interpretation (SLIPs versus limiting data)? Quite frankly, I would consider that more important than some of the panels in Figs. 2 and 3.

On key terminology, is Relative Water Level something different than the widely used Reference Water Level? Also, with respect to the RWL and IR calculations it would be helpful to clarify why MSL is used, as opposed to the commonly used MTL (e.g., Shennan, 1986). I partly bring these things up because I was struck by the fact that the indicative range for most of the sea-level indicators is the same, which raises the question to what extent they are mutually exclusive.

With regard to basal peat, it is fine to initially interpret this as a GWL indicator, but (1) it is unclear what the depth range information in line 497 is based on; and (2) by not including the next step (i.e., how basal peats can become SLIPs) it isn't clear how these could be anything more than limiting data.

The main element that is missing here is paleoenvironmental information (e.g., biological, geochemical) that is necessary to determine whether basal peat is intertidal. From this perspective, it is unfortunate that biological information ("marine fauna") is divorced from the other sea-level indicators. Absent this type of evidence, an interpretation where the initial stage of basal peat formation occurred in a fresh environment (lines 489-491) becomes rather speculative. Separate from this, note that basal peat is commonly defined as resting immediately on a largely compaction-free substrate, so other than thickness loss of the sampled interval, there would be no vertical displacement. For tops of peat beds this is of course different but referring to those as basal peats (lines 671-672) is bound to lead to confusion.

I was unfamiliar with the concept of estuarine terraces (Section 3.2.4). Elsewhere (line 672) I came across the term "estuarine tidal flat tops" and the spreadsheet mentions "preserved tidal flat surface." What is needed here, is a clear explanation of what this would constitute in a modern tidal/estuarine environment. Were these tidal flats? And if so, how exactly is it defined? Note that it is not uncommon to restrict tidal flats to mudflats that reside between MTL and MLW (or MLLW), i.e., excluding marsh platforms (e.g., Fagherazzi et al., 2006, PNAS). The formulas indicate otherwise, however, placing this category in the same elevation range as coastal marshes... so what is the difference? As mentioned before, a cartoon would be of great help here – it would enable the authors to more effectively communicate the relationships between the different sea-level indicators. As mentioned, the different categories must be mutually exclusive, but are they? Along similar lines, is there a difference between marine terraces and abrasion platforms (lines 1032 and 1034)?

With regard to elevation (Table 5 and associated text), cross sections are not a measurement technique, merely a means to illustrate data. What matters here is how elevations as shown in a cross section were determined. Also, what does "±10% of elevation measurement" mean? For example, if an elevation of 10 cm above MTL is obtained, the error would be 1 cm, but if it is 1 m, the error would be 10 cm? I doubt that this is what the authors meant. A larger question is whether elevations are related to modern MTL or simply to a geodetic datum. If the former, how do the authors account for the fact that MSL is subject to fairly rapid change, often of opposite sign as in Britain (lines 642-645)? Finally, vertical uncertainties of only ±1 m (line 1243) for the LIG seem unrealistically low. That would place them among the best RSL data even within much of the Holocene. I find it difficult to see how adding >100 kyr would not inevitably result in larger errors. Am I missing something?

***Reply (AC1 continued):***

- This comment was also made in RC2. We have revised the beginning of this section, including a new Figure 2.

- Regarding the importance of the section/figures relative to Research History section: this section has now been moved to an appendix.  Though this moves away from the structure of the special issue template, it is the shear wealth of >100 years of research in this area means it made the main text very long. As a result, the revised manuscript now focuses earlier the indicator types and database entry.

- RWL is now correctly de-abbreviated. Mutual exclusivity of the SLIP types is now addressed at the beginning of the describing section (using a figure and a decision tree annex to Table 3).

- Regarding particularities of recognising basal peats (in regional geological mapping, sedimentology) and working them up to basal-peat SLIP types (a later step of interpretation than the mapping): the section is edited, misrepresentations corrected. The listing in this paper does not intend to be a review of LIG SLIP indicator types and their Holocene parents/analogies. In our opinion, the basal peat

indicator type has been described before in Holocene sea level databases and we reference sources in the text.

- Regarding the 'marine fauna' indicator type: The revision makes clear (a decision tree added as Table 3b), that we used this general indicator as a fall back option. Other indicator types also make use of 'marine fauna' information as part of the argumentation, but they have more contextual information.

- Regarding 'estuarine terraces': a new introduced cartoon figure should now further clarify this type. Recognizing *Intertidal* sedimentological signature and associated biota is key for this particular one. Whether that is in muddy or sandy settings in not of prime importance. Entries in the spreadsheets echo descriptions in the original literature.

- Regarding 'elevation' accuracy: We have modified Table 5 to fix referring to reading from a graph as reporting a measurement technique. For some older sites only graphical presentation in publications remain and therefore 'read from a cross-section' becomes the way the elevation data was entered in the database and is used by other papers in the special issue. We will forward the particular comments, also to % inaccuracy, to the WALIS database interface (Rovere et al. 2020). WALIS stores two vertical uncertainties: those of the present day vertical position of the indicator (Table 5), and those of the palaeo sea-level elevation of the SLIP (using the 'in formula' part of the indicator description). Only in the case that also Vertical Land Motion is considered does the vertical uncertainty become age-attribution dependent. In a few cases (older literature) there are links between variable MTL and the expression of present day vertical position errors (often the same ones that are 'read from cross-section'). The setup of the WALIS is such that a tidal range has to be provided (upper and lower limit), separately from the vertical elevation. This is used in the calculation of the palaeo sea-level elevation and in the propagation of the uncertainty. A user can assess this on point-by-point basis, and decide to overrule deemed too small uncertainties. As for now, we have not modified our database entries in keeping with the WALIS structure.

**Reviewer Comment 1 (RC1) continued (3/3):**

[3] Portions of the text (notably Sections 2 and 5) tend to get lost in lengthy regional stratigraphic details, with associated nomenclature that many if not most readers will be unfamiliar with. For example, the authors repeatedly mention the Lusitanian (a chronostratigraphic unit?) that I couldn't find in Table 1. Lincolnshire is also mentioned multiple times but not shown on a map. Note that these are merely examples. Trimming down this section would make the paper much more accessible. Would it be possible to move most of the regionally specific text to a supplement?

I came across tons of grammatical errors. The nature of the work is such that accurate communication is essential. I know that this is more about bookkeeping than imagination (and thus not particularly exciting) but failing to do so might cause more confusion than resolution. Put differently, other authors should be able to reproduce the work. In its current form, the writing is quite rough, and at many occasions difficult to follow. For a small selection of examples, see lines 147, 213, 300-301, 334, 398-399, 435, 625, 727-728, and 1091. Without a doubt, an effort by all authors can address this, not least the native English speakers on the team.

*Reply:*

Yes, we agree. We have majorly edited the text and moved some sections to an appendix. Specific comments of RC1 regarding textual changes were implemented as part of this process (see below).

**RC1 line-by-line textual comments:**

These were implemented at the start of our revision (before the restructuring).

Line 55: I suggest adding Shennan (1982, Proc Geol Assoc) here. Also, if the journal guidelines allow it, I would recommend listing references chronologically.

DONE

Line 80: there is a widespread preference for "numerical ages" over "absolute ages" (see Colman et al., 1987, QR).

DONE (yes, you are right).

Line 99: not sure what "superregional" means.

EXPLAINED

Line 104: you probably mean "paleoceanographic" here.

DONE

Line 115: why not just stick to "sea-level data points" as in the title?

DONE (yes, you are right, this resulted from rearranging blocks of text earlier).

Line 125: better to say "unlike the present interglacial."

DONE, agreed.

Line 143: I suppose this must be cm?

NO, the units were correct, but confusion arose. We think this came from not mentioning the ballpark rates of differential VLM (0.1-0.3 m/kyr {=mm/yr cm/cy}). Sentences were adapted.

Lines 194-195: not clear what a "sea-level site" is.

DONE: REPHRASED

Line 290: you define Cyprina Clay here, but it has already been mentioned earlier (line 239) without a definition.

LEFT AS IS: Line 239 contained a cross-reference to section 2.4 (=line 290).

Line 308: here and elsewhere (wherever appropriate) I suggest you use "succession" rather than "sequence" unless it explicitly refers to sequence stratigraphy.

CONSIDERED BUT NOT FULLY TAKEN OVER. We use facies successions, but stratigraphic sequences. The term sequence is not exclusive to sequence stratigraphy. We reduced reference to sequence stratigraphy from the beginning of the manuscript onwards (see above) and think we have neutralized the interpretation that using the word sequence can trigger that way.

Line 313: please try to be more precise with terminology. Note that littoral environments occur in lakes as well, and lacustrine is the more commonly used term rather than limnic. Why not just referring to this as lacustrine versus marine (or shallow marine)?

DONE. Lacustrine replaced with limnic. Baltic Sea is notorious for transitional brackish standing water phases with littoral deposits continued to be produced, transitional of being lacustrine or marine.

Line 447: here and elsewhere, the authors use the term "generic" (or "non-generic") but it isn't particularly clear what this means.

DONE. We reduced the use of the word. It is now only used in relation to Table 2, where it is explained. 'WALIS-generic'.

Line 451: "subaquatic" should be "subaqueous."

DONE

Lines 461-465: what is said here about tidal amplification/dissipation matters a great deal for sea-level reconstruction and needs to be supported by one or more references.

DONE. The sentence was modified. A reference was added.

Line 481: the definition that is provided here is debatable; basal peats typically overlie the transgressive surface that represents the upper boundary of the lowstand (or falling stage) systems tract (e.g., Zaitlin et al., 1994, SEPM Spec Pub).

LEFT AS IS, position clarified in text and new Figure 2b. Two ways of looking at it exist: the one quoted by the reviewer, and the alternative that we took up. In regional-scale (basin-scale) 'sequence stratigraphy', the TOP of the basal peat is the transgressive contact. The basal peat is the last terrestrial deposit, the deposit overlying it is the first subaquous deposit. This follows Catuneanu (2006) and Hijma & Cohen (2011). The lowstand stage systems tract extends all the way up to the transgressive contact in this definition. The transgressive contact is diachronous too in this definition.

Lines 485-487: the paleosol interpretation is tenuous, especially in the absence of references. The evidence I'm familiar with (e.g., Vetter et al., 2017, G3) shows that paleosols formed immediately prior to basal peat accumulation. Or are the authors referring to entirely different paleosols? Please clarify.

CLARIFIED in text, and in Figure 2b. Relatively immature paleosols (young floodplains) underly some basal peats (the ones over valley floors). More mature paleosols underly some other basal peats (the ones formed outside drowned valleys).

Line 571: the RWL should be defined more precisely; I presume it is this range divided by two?

DONE. We echo the formulations of the WALIS interface in the SLIP type definition text.

Line 597: "footer" should be "footnote."

DONE

Line 620: is the next section really about the English Channel? It certainly doesn't appear to be about chronostratigraphy?

DONE (removed)

Lines 634-636: are you sure "subtract" is correct here? Shouldn't this number be added to bring an elevation from low tide level to mean tide level?

DONE. It was correct as written. The datum is raised by +2.33m. Values reported to original TAW have to be lowered by 2.33m. A value reported as 0 m TAW equals -2.33 m TAW+2.33. We edited the sentence and added that calculation example. IN FURTHER REVISION: WE DECIDED TO DROP THIS SECTION.

Lines 665-671: Keogh et al. (2021, JGR-ES) discuss the time-dependent nature of compaction of organic-rich coastal deposits, including its implications for sea-level reconstruction.

Thank you for the reference. After checking it, we have added it to the citations.

Line 678: is the nature and thickness of overburden included in the database?

YES, described in note fields for those entries where a compaction assessment was incorporated.

Line 696: note that VLM strictly includes compaction, but that is clearly not how the authors define things in this work. This needs to be clarified somewhere.

VLM in WALIS' context is a linear term: a rate that multiplied by age gives a vertical correction. We use it to specify corrections for North Sea basin subsidence (which in turn has several components, e.g. Kooi et al., 1998). In WALIS this linear term is applied independently of compaction corrections.

Lines 701-703: I have trouble following this sentence; what does "earlier applied basin subsidence" mean?

ACTED. This part of the text was edited

Lines 731-733: using uncertainties is good, but it raises the question why no uncertainties are used for areas outside of the North Sea Basin where VLM appears to be less well understood (also see my earlier comment about vertical uncertainties). This must be addressed.

ACTED UPON. FIRSTLY, IN THE REVISED TEXT for the North Sea we use the wording "we develop updated VLM corrections based on geological information independent of the LIG datapoints", whereas "along the English Channel. Independent

estimation of the rates of VLM are not available, as studies that provided uplift rates did so based upon the same marine terrace elevations that we consider our SLIPs." SECONDLY, in the closing remarks we added a recommendation paragraph.

Reasons why we did report further vertical uncertainties for the English Channel, but do so for the North Sea basin are related to the setup of the WALIS database: (1) In its design it gives component uncertainties and calculates a non-VLM corrected palaeosea-level elevation (including user specified compaction corrections, and propagating specified uncertainties), but leaves it to the user to choose to apply a VLM value. (2) For the North Sea we included VLM specification, because earlier literature had also done this (for the 9 Zagwijn-1983 data points) and one wishing to use all ~50 data points from the basin subsidence affected area, is offered opportunity to consistently do so. In the paper we also advocate to apply it (e.g. in Figure 5/now Figure 4), and we verified if we provided (reproduced) more or less the same values as were also used before (Lambeck et al., 2006; Kopp et al. 2009; also Kiden et al., 2002 and Vink et al., 2007). (3) For the English Channel and British side of the North Sea (and also the Baltic Sea side of Denmark and Germany): we agree that ideally one should also apply VLM correction and associated uncertainty correction to areas outside the North Sea Basin. However, no generally accepted reference for this is known to us. Here dependent/independent becomes a thing: Along the English Channel, total uplift is inferred from geomorphological features (terrace staircases) and subsequently attributed to causes (part tectonic, part glacio-hydro-isostatic). Also, it has often used assumptions on sea-level high stands as input. An outcome of the interdependencies, is that different authors have put different emphasis on root causes. We therefore report the total uplift rates from our review, and indicate (in the Closing Remarks paragraph) that these are at best maximum VLM rates for this area, and that further work is needed.

Line 1202: expressions like "5* LIG SLIPs" are not really appropriate for a paper like this one. If there is a ranking of data point quality, this should be clarified.

Removed as part of the editing. There is indeed a star ranking system in the database design, but we do not discuss it in this paper.

Lines 1231-1233: the authors seemingly intend to make a significant point here, but I had trouble understanding what that point is. Are they highlighting the differences or the similarities between WALIS and Holocene RSL databases?

We appreciated this remark. The Closing Remarks subheading originated from the Special Issue template. We added a short paragraph that echoes of matters of LIG-Holocene crosscomparison touched upon in other sections of the paper.

Line 1281: this link took me to a southern African database. I suppose something else was intended here?

FIXED, apologies.

The quality of the figures needs work; for example, the lower portions of Figs. 5 and 6 are very difficult to read. A few more specific suggestions regarding figures and tables:

Fig. 1: the differentiation between small and large rivers is not meaningful (I can't tell the difference in the map). The depocenter VLM information from Fig. 4 shouldn't be repeated here. Instead, make sure that all geographic names mentioned in the text can be found in the maps.

IMPLEMENTED. 'all' geographic names mentioned in the text was not possible graphically.

Fig. 2 is not of the greatest quality and is every panel referred to in the text?

The figure is moved to the appendix

Fig. 3: please show the location of these areas in Fig. 1.

The figure is moved to the appendix. We did not add Appendix figure boxes in Fig. 1.

Table 1: Last Interglacial covers all of MIS 5, which seems inconsistent with the LIG definition used throughout the paper (MIS 5e or Eemian; also see line 100).

The table was modified. With hindsight, Table 1 drew the wrong attention (too much on worldwide, and with errors). We considers a major flaw in our first submission.

Table 2 needs a lot of attention; it contains a ton of text, and numerous acronyms that are not explained (note that it should be possible to read the table content without having to consult the main text). It would probably help to simplify things a bit. For example, could a finite list of geomorphic/stratigraphic contexts be provided and show which ones occur in a given geographic region? Maybe the table is simply too large, and this aspect should be separated from the numbers of data points in each geographic region.

The table was simplified. It now fits on one page. The column 'Quaternary Terrain' was deleted.

Table 3: within this table there is a header that says "Paper Author added as part of this study." I had to read this several times, but I'm still not sure what it means. Also, please specify which other WALIS Special Issue papers are being referenced here.

The table was updated. There is now also a Table 3b as part of the revision.
Two specific papers for the SI are now referenced. There are more… the SLIP types are quite widely used.

Table 4: the title is awkward; please reword. And what exactly is meant with "Duration constraint"? Is this the maximum duration? I looked at the various numbers in the table but couldn't make sense of this.

The table header was updated. The table contents was also updated. The Duration header was updated. It is a minimum duration based on varve-counts and conservative extrapolations thereof.
* * *
**Reviewer Comment 2 (RC2) – (1/2):**

The authors evaluate the published Last Interglacial sea-level data points along coastlines adjacent to North Sea and English Channel from northwestern Europe, and compile the Last Interglacial sea-level data set in detail. The authors also point out the importance of the NW Europe sea-level data set, which reflects MIS 6 ice sheet history, and the difficulties and suggestions for the further LIG sea-level researches in NW Europe (including the geochronology issues). Thus, this is a very valuable compilation of the Last Interglacial sea-level data.

I have two comments for this manuscript. Firstly, the RWLs and IRs for the sea-level indicators in section 3.2 needs to be well-defined in the paragraphs. I cannot fully understand how the RWLs and IRs of each indicator are decided in the 'In formula' paragraphs. Since these sea-level indicators are rare outside NW Europe, some figures and more detail information for these sea-level indicators are necessary.

*Reply:*

\* This comment is also made in RC1 (their second comment).

\* Suggestion of a figure illustrating the SLIP types that outside NW Europe, in LIG contexts are rare: such a figure has been added (new Fig. 2 in revised manuscript).

\* RWL and IRs for sea-level indicator requiring explanation: the text on this has been edited. Closing lines 'as formulas' left as is, because specific to the WALIS database setup, and the concepts behind it are covered elsewhere (Rovere et al., 2016; 2020)

**Reviewer Comment 2 (RC2) – (2/2):**

Secondly, there are lots of editing issues in this manuscript, including inconsistent word spellings and typos (listed below). These editing issues make this manuscript sloppy and cannot be read smoothly.

*Reply:*

\* We agreed and have acted. All specific comments of RC2 were implemented as part of the major revision (see below).

**RC2 line-by-line textual comments:**

Line 3: Department 'of' Physical Geography?

DONE

Line 16, 18: is it 'highstand', 'high stand', or 'high-stand'?

DONE. Highstand all over, but we reduced greatly the use of the term for other reasons (addressing RC1 comment)

Line 25: 'morpho-stratigraphic' or 'morphostratigraphic'?

DONE. morphostratigraphic

Line 41, 53, 287, 396, 924: 'palaeoenvironment' or 'paleoenvironment'?

DONE. Palaeo

Line 46, 435, 452, 454, 484, 668, 699, 700, 930: is it 'meter', 'decimeter', or 'metre', 'decimetre'?

DONE. Meter

Line 64, 139, 156, 435, 642, 697, 700, 728, 731, 733: are they 'center', 'depocenter', or 'centre', 'depocentre'?

DONE. Centre

Line 87, 180, 902, 925, 1050, 1212, 1219, 1250: are they 'MIS 3, MIS 5e, MIS 6, and MIS 7', or MIS-3, MIS-5e, MIS-6, and MIS-7'?

DONE

Line 95, 158, 159, 207, 296, 305, 325, 334, 342, 361, 555, 558, 569, 923, 931, 937, 947, 951, 954, 959: 'mollusc', 'mollusca', or 'molluscan'?

CHECKED. Improved. Mollusca, not molluscs.

Line 151, 755, 773, 790, 836, 846, 1237, 1251, 1261: 'datapoints' or 'data points'?

DONE, data points

Line 166-172: Follow the rule of the journal. Use lower case in figure captions (a-e) rather than initial characters (A-E).

DONE. Figure 2 was revised. It was also moved to the Appendix, where it is Fig. A1.

Line 169 and 171: Eemian 'Highstand' or 'highstand'?

DONE highstand, no caps

Line 207: what is 'resp.'?

DONE respectively

Line 217: 'low stand' or 'lowstand'?

DONE lowstand

Line 296: 'Cyprina Clay' or 'Cyprina-Clay'?

DONE

Line 308: 'would be' or 'would-be'?

(section was edited later on)

Line 391: the abbreviation of thermoluminescence (TL) and optically stimulated luminescene (OSL) before this line (e.g. line 80). Should it be mentioned in the earlier section?

(section was edited later on)

Line 417: 'Chalk' or 'chalk'?

DONE Changed to chalk.

Line 457: Hijma and Cohen (2011; 2019); Peeters et al. (2016; 2019)

DONE

Line 463: 'HAT' is mentioned here first so please express it in full term.

Edited

Line 477: 'the' rather than 'The'.

Left

Line 700: is there any reference for the rates of VLM?

YES. It is Kooi et al. 1998, updated). We include this reference now. The sections were further edited.

Line 721: 'characterizing' or 'characterising'?

DONE characterising

Line 786: 'RSL' or 'SLR'?

DONE. SLR spelled out (sea-level rise). RSL used a lot through the text (relative sea level).

Line 1099: where is Fig. 7?

FIXED. Reference removed. There is no Fig. 7. We once considered to produce it, but dropped this prior to submission.

Line 1126 and 1131: 'litho-stratigraphic' or 'lithostratigraphic'?

DONE. lithostratigraphic

Line 1202: 5*?

SEE REMARK IN RC1. Five-star as in a rating. This is edited out now.

Line 1270: (!)?

DONE. Removed.

Figure 1: 'depocenter' or 'depocentre'?

Depocentre -- FIXED IN CAPTION.

Figure 2: The citation is not consistent: Zagwijn-83, dM&dB 1973

DONE. Figure 2 was revised, and it was moved to the Appendix, where it is Fig. A1.

Figure 4: several numbers on the figure are overlapped.

This is fixed (and Figure 4 is now Figure 3)

Table 1: editing issues such as 'w large oscillations' and 'Weichselia' and 'n'.

TABLE 1 WAS REVISED. Formatting issues resolved with that.

Table 2: 'MIS 6' or 'MIS-6'? 'Chalk' or 'chalk'?

This column in Table 2 has been removed as part of the revision.